# Wasserstein Bounds for generative diffusion models with Gaussian tail targets

**Xixian Wang**                                                              *xixian001@e.ntu.edu.sg*
*Division of Mathematical Sciences, School of Physical and Mathematical Sciences*
*Nanyang Technological University*
*Singapore 637371*

**Zhongjian Wang**\*                                                         *zhongjian.wang@ntu.edu.sg*
*Division of Mathematical Sciences, School of Physical and Mathematical Sciences*
*Nanyang Technological University*
*Singapore 637371*

**Reviewed on OpenReview:** *https://openreview.net/forum?id=QbQ4DtP5vS*

## Abstract

We present an estimate of the Wasserstein distance between the data distribution and the generation of score-based generative models. The sampling complexity with respect to dimension is $\mathcal{O}(\sqrt{d})$, with a logarithmic constant. In the analysis, we assume a Gaussian-type tail behavior of the data distribution and an $\epsilon$-accurate approximation of the score. Such a Gaussian tail assumption is general, as it accommodates practical target distributions derived from early stopping techniques with bounded support.

The crux of the analysis lies in the global Lipschitz bound of the score, which is shown from the Gaussian tail assumption by a dimension-independent estimate of the heat kernel. Consequently, our complexity bound scales linearly (up to a logarithmic constant) with the square root of the trace of the covariance operator, which relates to the invariant distribution of the forward process.

## 1 Introduction

Diffusion models (DM) are among the state-of-the-art tools in the new GenAI era. As generative models, diffusion models first link the target distribution to a distribution that is easy to sample via a diffusive process (forward). The generative (backward) process then *reverses* the diffusion, enabling samples from the easily sampled distribution to be transformed into samples from the target distribution. A well-known mathematical model that encapsulates this approach is the score-based stochastic differential equation (SDE) model (Song et al., 2021), where the forward and backward processes are represented by two SDEs that share the same marginal distribution (Anderson, 1982; Haussmann & Pardoux, 1986). In most cases, the forward process is assumed to be an Ornstein-Uhlenbeck (OU) process. The backward process incorporates the gradient of the logarithmic density (score) of the forward process. When the explicit form of the score is unknown, it is estimated by a neural network from discrete samples of the target distribution.

A major direction for the theoretical study of DMs is the convergence of the approximated backward process with limited data assumptions. Progress in this direction has been made, for example, in works such as (De Bortoli, 2022; Chen et al., 2023a; Benton et al., 2024; Conforti et al., 2024; Gao et al., 2025; Li & Yan, 2025; Silveri & Ocello, 2025; Beyler & Bach, 2025). Essentially, there are two camps regarding the availability of the Lipschitz bound on the score function. When the Lipschitz bound of the score is available (regular target), the backward process is generally well-defined until $t = 0$, and convergence results are related to the Lipschitz bound, see e.g., (Chen et al., 2023a; Conforti et al., 2024; Gao et al., 2025). Otherwise, when the

---

\*Corresponding author. Address: 21 Nanyang Link, SPMS-05-05, 637371, Singapore. Email: zhongjian.wang@ntu.edu.sg.

Table 1: Summary of previous and our results for score-based diffusion models in $d$ dimensions. The complexity bound gives the number of steps $N$ needed to ensure error $\leq \epsilon_0$.

| Target $\overrightarrow{P}_0$ | Metric | Complexity | Result |
|---|---|---|---|
| $\mathbb{E}|X_0|^2 < \infty$ | $\mathrm{TV}(\overrightarrow{P}_0||\overleftarrow{Q}_T)$ | $N = \mathcal{O}\left(\frac{d}{\epsilon_0}\right)$ | (Li & Yan, 2025) Thm. 1 |
| $\nabla \log p_0$ $L$-lip | $\mathrm{KL}(\overrightarrow{P}_0||\overleftarrow{Q}_T)$ | $N = \mathcal{O}\left(\frac{d^2}{\epsilon_0}\right)$ | (Chen et al., 2023a) Thm. 5 |
| $\mathcal{F}(\overrightarrow{P}_0|\mathcal{N}(0,I_d)) \lesssim d$ (*) | $\mathrm{KL}(\overrightarrow{P}_0||\overleftarrow{Q}_T)$ | $N = \mathcal{O}\left(\frac{d}{\epsilon_0^2} \log \frac{d}{\epsilon_0}\right)$ | (Conforti et al., 2024) Thm. 1 |
| $P_0$ strongly log-concave+ $L_0$-smooth | $\mathcal{W}_2(\overrightarrow{P}_0, \overleftarrow{Q}_T)$ | $N = \mathcal{O}\left(\frac{d}{\epsilon_0^2} \log \frac{d}{\epsilon_0}\right)$ | (Gao et al., 2025) Tab. 2 |
| $P_0$ one-side Lipschitz + weakly log-concave | $\mathcal{W}_2(\overrightarrow{P}_0, \overleftarrow{Q}_T)$ | $N = \mathcal{O}\left(\frac{d^2}{\epsilon^2}\right)$ | (Silveri & Ocello, 2025) Thm 3.5 |
| G tail, Ass. 2 | $\mathcal{W}_2(\overrightarrow{P}_0, \overleftarrow{Q}_T)$ | $N = \mathcal{O}\left(\frac{\sqrt{d}}{\epsilon_0} \log \frac{d}{\epsilon_0^2}\right)$ | This work: Cor. 3.5 |
| $\mathbb{E}|X_0|^2 < \infty$ | $\mathrm{KL}(\overrightarrow{P}_\delta||\overleftarrow{Q}_{T-\delta})^{(**)}$ | $N = \mathcal{O}\left(\frac{d\log^2 \frac{1}{\delta}}{\epsilon_0}\right)$ | (Benton et al., 2024) Cor. 1 |

\* $\mathcal{F}$ denotes relative Fisher information. Our Gaussian tail assumption (Assumption 2) implies $\mathcal{F}(\overrightarrow{P}_0|\mathcal{N}(0, I_d)) \lesssim d$, and they are equivalent under the standard Gaussian case.
\*\*$\delta$ denotes the early stopping time, Benton et al. (2024) compares the KL-divergence between the sampled distribution and the early stopped data distribution as the original data distribution under their assumption may be non-smooth.

bound is unavailable (singular target), the early stopping technique is introduced, and convergence results are related to the stopping time (De Bortoli, 2022; Lyu et al., 2022; Beyler & Bach, 2025). Various analytical approaches have been adapted for these two types of assumptions.

**In this work**, we aim to present a general error analysis that applies to both regular and singular target distributions and also generalizes to an infinite-dimensional setting. The analysis is based on *a dimensionless and global-in-space Lipschitz bound* of the score, derived from a heat kernel estimation approach. Our complexity bounds, derived under the *Wasserstein-2-metric*, are *linear (with a logarithmic constant) in the square root of variance* of the Brownian motion in the forward process (Theorem 3.4). In the finite-dimensional case with the standard Gaussian as the base distribution (the invariant measure of the forward process), the variance is linear in the dimension and hence the complexity is $\mathcal{O}(\sqrt{d})$. For the infinite-dimensional case, a Gaussian random field is taken as the base distribution, and the variance is linear with the trace of the covariance operator.

It is worth noting that the results of this work adopt the *Wasserstein-2 distance* as the metric of error (instead of Kullback–Leibler (KL) divergence) due to its flexibility. More precisely, the reasons are two-fold: **(1)** In practical applications of diffusion models for the generation of structured data (image, text, video, protein, etc.), the target distributions mostly find their support in a compact sub-manifold, see further discussion of the manifold hypothesis in (Tenenbaum et al., 2000; Bengio et al., 2013; 2017). Consequently, under this hypothesis, the standard KL divergence cannot be consistently defined between the distribution obtained via the backward process, whose support is the entire space, and the target distribution with compact support. **(2)** In high (towards infinite) dimension settings, it becomes necessary to compactify the forward process. One way to achieve this is by choosing the covariance matrix (operator) $C$ in the forward process (1) to have finite trace, so that the invariant distribution of the process has a finite second moment. The Wasserstein distance then scales with $\mathrm{Tr}(C)$, making it consistent with this compactification in the context of infinite-dimensional generative models. By contrast, the KL divergence scales with the ambient dimension and therefore cannot be directly extended to yield a dimensionless result. A motivating example is provided in Appendix B.1.

**Related work**

Complexity bounds (De Bortoli, 2022) established the first convergence guarantees in the 1-Wasserstein distance, assuming that the data distribution satisfies the so-called manifold hypothesis. Some recent works (Pierret & Galerne, 2024; Gao & Zhu, 2024; Gao et al., 2025; Silveri & Ocello, 2025; Li & Yan, 2025; Bruno & Sabanis, 2025; Beyler & Bach, 2025) established convergence guarantees under Gaussian, log-concave, weakly log-concave, or other minimal assumptions. We are also aware of several complexity bounds under KL divergence, for instance (Chen et al., 2023a; Benton et al., 2024; Conforti et al., 2024). A common point of these approaches is the use of the chain rule for KL divergence, which follows from the Girsanov theorem, to

separate the global error into local truncation errors. To bound local ones, the probabilistic viewpoint, which estimates the score function under the distribution of the forward process, comes into play. In this work, we instead provide a point-wise estimate of the score. Such an estimate also facilitates the analysis of the score under the approximated backward process, thereby providing a Wasserstein bound. In Table 1, we present the comparison of the complexity bounds.

Contractivity of the dynamics A common technical difficulty in establishing convergence guarantees under the $W_2$ metric is that the reverse-time dynamics typically needs to exhibit a form of contractivity. Indeed, Wasserstein-based analyses rely on controlling the propagation of perturbations along the backward process, which becomes tractable when the reverse-time drift is contractive; see, for example, the technical overview in (Li & Yan, 2025).

For this reason, many earlier works enforce contractivity through strong structural assumptions on the data distribution, such as strong log-concavity or closely related regularity conditions. These assumptions ensure global contractivity or stability of the reverse-time dynamics, leading to $W_2$ convergence guarantees; see, for instance, (Gao et al., 2025; Bruno et al., 2025; Yu & Yu, 2025; Tang & Zhao, 2024; Strasman et al., 2025). Subsequent works have progressively relaxed these requirements, considering weaker notions such as weak log-concavity combined with one-sided Lipschitz conditions, or semiconvex target distributions with potentially discontinuous gradients; see, e.g., (Silveri & Ocello, 2025; Bruno & Sabanis, 2025). In contrast, for the KL divergence, there are works that obtain convergence guarantees with polynomial-type dependence on the problem parameters without assuming strong log-concavity or global contractivity of the reverse-time dynamics; see, for instance, (Chen et al., 2023a; Benton et al., 2024; Conforti et al., 2024).

Rather than enforcing global contractivity of the reverse-time drift via structural assumptions on the target distribution, this work takes a different approach. We exploit the regularizing effect of the forward Ornstein–Uhlenbeck diffusion. In particular, although the reverse-time process may not be uniformly contractive, the non-contractive component of the drift admits a Lipschitz bound that decays exponentially in time (see Corollary 3.2). As a result, the influence of this component diminishes over the diffusion time horizon, yielding a stabilization sufficient for convergence under the Wasserstein metric.

Diffusion models in infinite dimension While diffusion models are usually defined in finite-dimensional spaces, many applications involve probing infinite-dimensional distributions. For instance, in some Bayesian inverse problems (Stuart, 2010), one requires sampling the posterior distribution of a continuous function from its discrete observation data. Up to now, there are many studies applying diffusion models to functional spaces (Lim et al., 2025; Kerrigan et al., 2023; Pidstrigach et al., 2024; Franzese et al., 2024). While many theoretical studies (De Bortoli, 2022; Chen et al., 2023a; Benton et al., 2024) suggest that performance guarantees deteriorate with increasing dimension. (Pidstrigach et al., 2024) established a bound on the Wasserstein-2 Distance from the samples to the target distribution, which is dimension-independent but grows exponentially with the running time $T$. In this work, we further provide a Wasserstein-2-bound that is both dimension-independent and running time uniform under the Gaussian tail assumption.

Lipschitz bounds of the score The score functions in the SGMs are related to the gradient of log-density $(\log p)$ of the forward process. It is well known that the function $\log p$ itself follows a viscous Hamilton-Jacobi (vHJ) equation (Øksendal, 2003; Evans, 2022), as seen in (8) in the later discussion. Then the Lipschitz bounds of the scores are equivalent to Hessian bounds for a vHJ equation. There are various regularity results in the literature for the original Fokker-Planck equation or the transformed vHJ, see (Fujita et al., 2006; Strömberg, 2010) and recent results in (Blessing & Kupper, 2022; Mooney et al., 2025). We would point out that, except (Mooney et al., 2025), most results are seeking a spatially global Hessian bound (Chen et al., 2023a;b) which only lasts for a finite time without the Gaussian tail assumption (Assumption 2) in this work. (Mooney et al., 2025) also provides a local-in-space and global-in-time bound, while only polynomial in dimension $(d^3)$ complexity can be derived from it due to the spatial locality.

We are also aware of the literature on the topic of contraction properties or Lipschitz estimates of transport maps between measures.[1] The *Caffarelli's contraction theorem* (Caffarelli, 2000) in the optimal transport (OT) setup is the starting point, and see (Colombo et al., 2015) for generalization to Lipschitz estimate.

---

[1]More precisely, in the context of this work, the transport map refers to the push-forward map from the Gaussian (base distribution) to the target distribution $p_0$.

Besides OT, the contractive map can also be attained by (reverse) heat flow (Kim & Milman, 2012). Recent generalizations to the Lipschitz estimates of the maps, including (Mikulincer & Shenfeld, 2023; Neeman, 2022; Fathi et al., 2024; Brigati & Pedrotti, 2025), obtain results for the boundedness of flow in a similar fashion to this work under diverse assumptions. In comparison, this work focuses more on deriving theories with applications to score-based diffusion models. In this regard, our assumptions cover the spatially anisotropic noise (no necessity for equivalence among $A$, $C$, and $I_d$ in Assumption 2) and we discuss the convergence and complexity bounds in the discrete approximation of the generative flows.

**The main contributions of this paper are:**

- We introduce a Gaussian tail assumption (Assumption 2) and apply a heat kernel estimate (Theorem 3.1) inspired by the solution of the viscous Hamilton–Jacobi (vHJ) equation to obtain a spatially global Lipschitz constant for the modified score function that decays exponentially over time (Corollary 3.2). This assumption accommodates non-log-concave target distributions and covers practical scenarios, such as when the support of the target distribution is bounded and the early stopping technique (Lyu et al., 2022) is employed (Theorem 3.8).
- We establish a Wasserstein-2 bound (Theorem 3.4) that depends only on the second moment of the base distribution, allowing the result to extend naturally to infinite-dimensional settings. When the second moment scales proportionally with the ambient dimension, the resulting sampling complexity achieves the state-of-the-art dependence on the dimension, with a rate of $\mathcal{O}(\sqrt{d})$. (Gao et al., 2025) Proposition 6 also shows that under the standard Gaussian distribution, such a complexity bound is optimal.
- As part of our methodological contribution, we provide a modified approach rather than conventional Lyapunov-type analysis, which utilizes the exponential decay structure of the score to show the uniform boundedness of accumulation error for an arbitrary long forward/backward process.

## 2 Preliminaries

This section contains the necessary notation and background to set the stage for the rest of the paper. We begin by introducing basic notations used throughout the text, followed by a review of the forward and backward processes in continuous time. We then discuss the discrete-time approximation commonly employed in diffusion models, together with its practical relevance in training score-based models. Finally, we return to the continuous-time setting to present the viscous Hamilton–Jacobi (vHJ) formulation that lies at the heart of our regularity analysis.

**General Notations**  Let $\gamma_C$ be the Gaussian measure $\mathcal{N}(0, C)$. For an $n \times n$ matrix $A$, we use the operator norm $\|\cdot\|$,

$$\|A\| = \sup_{v \neq 0} \frac{|Av|}{|v|} := \text{the largest eigenvalue of } \sqrt{A^T A}.$$

For a symmetric, positive-definite $n \times n$ matrix $A$, we use $|\cdot|_A$ to denote weighted $l_2$ norm in $\mathbb{R}^n$ such that,

$$|x|_A^2 := \langle A^{-1/2}x, A^{-1/2}x \rangle.$$

When $A$ is the identity matrix, we neglect the letter for simplicity and $|\cdot|$ is the standard $l_2$ norm in $\mathbb{R}^n$. For vector (matrix, correspondingly) valued function $f$ with $x$ as variable, $|f|_\infty := \sup_x |f(x)|$ (correspondingly, $\|f\|_\infty := \sup_x \|f(x)\|$).

### 2.1 Continuous-time formulation of diffusion models

A large class of generative diffusion models can be analyzed under the SDE framework (Song et al., 2021), which contains two processes: forward and backward. The forward process, which gradually transforms the data distribution into white noise, is an OU process as follows,

$$d\vec{X}_t = -\frac{1}{2}\vec{X}_t dt + \sqrt{C}dB_t, \quad 0 \leq t \leq T. \tag{1}$$

where $B_t$ is a standard Brownian motion, $C$ is a symmetric, positive-definite covariance matrix and $T$ is the final time such that the distribution of $X_T$ is close to the Gaussian distribution $\mathcal{N}(0, C)$, denoted as *base distribution*. The initial $\overrightarrow{X}_0$ follows the target(data) distribution, denoted as $\overrightarrow{P}_0$. We remark that in the majority of theories and applications of the diffusion model, $C$ is assumed to be the identity. We maintain the spatially anisotropic noise assumption ($C \not\equiv I_d$) in the following derivation to enable our theories to generalize to an infinite-dimensional setting where some compactification is necessary, see also (Pidstrigach et al., 2024). For compactness of the process, we made the following assumption to bound the second moment of the forward process.

**Assumption 1.** *The data distribution has a bounded second moment, $M_2 := \mathbb{E}_{\overrightarrow{P}_0}|x|^2 < \infty$. The covariance $C$ in* (1) *is in trace-class. Furthermore, we denote,*

$$M_0 = \max\{\mathrm{Tr}(C), M_2, 1\}. \tag{2}$$

We denote the probability density of the forward process $\overrightarrow{X}_t$ by $p_t$, then $p_t$ solves the Fokker-Planck equation with Cauchy data $p_0$:

$$\begin{cases} \partial_t p = \frac{1}{2}\big(\nabla \cdot (xp) + \nabla \cdot C\nabla p\big), \\ p(0, x) = p_0(x). \end{cases} \tag{3}$$

With time reversal $\overleftarrow{X}_t := \overrightarrow{X}_{T-t}$, the backward process $(\overleftarrow{X}_t)_{0 \le t \le T}$ satisfies the following SDE (Haussmann & Pardoux, 1986),

$$d\overleftarrow{X}_t = \big(\frac{1}{2}\overleftarrow{X}_t + s(T - t, \overleftarrow{X}_t)\big)dt + \sqrt{C}d\tilde{B}_t, \tag{4}$$

where $\tilde{B}_t$ is also a standard Brownian motion (may not be the same as $B_t$) and the term $s(t, x) := C\nabla \log p_t(x)$ is generally referred to as the score function. The process $(\overleftarrow{X}_t)_{0 \le t \le T}$ transforms noise into samples follows $\overrightarrow{P}_0$. We denote $\overrightarrow{P}_t$(correspondingly $\overleftarrow{P}_t$) as the marginal distribution of $\overrightarrow{X}_t$ in (1) ($\overleftarrow{X}_t$ in (4)). Then, $\forall 0 \le t \le T$, $\overleftarrow{P}_t = \overrightarrow{P}_{T-t}$, especially $\overleftarrow{P}_T = \overrightarrow{P}_0$.

## 2.2 Training and discrete-time approximation of diffusion models

Since the closed form expression of $p_0$ is unknown, the score function $s(t, x) = C\nabla \log p_t(x)$ is not available. Thus, we model the score function by a neural network $s_\theta(t, x)$, where $\theta$ denotes latent variables of the neural network. We train the network by optimizing an $L_2$ estimation loss,

$$\mathbb{E}_{\overrightarrow{P}_t}\|s_\theta(t, x) - C\nabla \log p_t(x)\|^2.$$

Given the estimated score $s_\theta$ (assumed to be $\epsilon$-accurate, specified in Assumption 3), one can generate samples of the target distribution $p_0$ by a numerical approximation of the backward process starting from the Gaussian distribution $\mathcal{N}(0, C)$,

$$d\overleftarrow{Y}_t = \big(\frac{1}{2}\overleftarrow{Y}_t + s_\theta(T - t, \overleftarrow{Y}_t)\big)dt + \sqrt{C}d\bar{B}_t. \tag{5}$$

In practice, diffusion models are trained and sampled in discrete time. Here, we introduce an Euler-type discretization of the continuous-time stochastic process, which facilitates the convergence analysis. Let $0 = t_0 \le t_1 \le \cdots \le t_N = T - \delta$ be the time discretization points (schedule), $\delta = 0$ for the normal setting and $\delta > 0$ for the early-stopping setting (Lyu et al., 2022), and we adopt the following discrete scheme. Starting from $\overleftarrow{Y}_0 \sim \mathcal{N}(0, C)$, for all $k = 0, 1, ..., N - 1$,

$$\overleftarrow{Y}_{t_{k+1}} = \frac{1}{\sqrt{\alpha_k}}\left(\overleftarrow{Y}_{t_k} + (1 - \alpha_k)s_\theta(T - t_k, \overleftarrow{Y}_{t_k})\right) + \sqrt{1 - \alpha_k}\bar{z}_k, \tag{6}$$

where $\alpha_k = \exp(t_k - t_{k+1})$ and $\bar{z}_k \sim \mathcal{N}(0, C)$ are i.i.d.

*Remark* 2.1. The discretization (6) approximately corresponds to the discrete-time scheme introduced in (Ho et al., 2020); it can also be found in the analysis work (De Bortoli, 2022).

The bounds established later (Theorem 3.4) quantify how well the discrete process approximates the continuous-time backward SDE in Wasserstein distance. The step size $\tau := \max_k(t_{k+1} - t_k)$ enters the final error bound in the form $\mathrm{Tr}(C)\tau^2$, demonstrating the accumulation of numerical error in the generative process.

### 2.3  Foundational Ideas based on Heat Kernel Estimation

The convergence analysis of the discrete scheme (6) relies heavily on the regularity (up to second order) of the score potential function $\log p_t$, which follows a viscous Hamilton-Jacobi equation (vHJ). While such derivations are known to experts (Evans, 2022), we briefly list the conversion between vHJ and heat equation for completeness.

We first consider the following transform of $\log p_t$,

$$q(t,x) = -\log p_t(x) - \frac{x^T \bar{A}_t^{-1} x}{2}, \tag{7}$$

where $\bar{A}_t = Ae^{-t} + C(1 - e^{-t})$. Then, $q$ satisfies the vHJ,

$$\begin{cases} \partial_t q - \frac{1}{2}\nabla \cdot C\nabla q + \frac{1}{2}|\sqrt{C}\nabla q|^2 + (C\bar{A}_t^{-1} - \frac{1}{2}I_d)x \cdot \nabla q = \frac{1}{2}\mathrm{Tr}(C\bar{A}_t^{-1} - I_d), \\ q(0,x) = -h(x), \end{cases} \tag{8}$$

where $h(x) := \log p_0(x) + \frac{|x|_A^2}{2}$ is the non-Gaussian part of the log-likelihood function (the remainder term in later Assumption 2). To simplify (8), we let $f(t) = -\frac{1}{2}\int_0^t \mathrm{Tr}(C\bar{A}_s^{-1} - I_d)ds$ and make a two step change of variables: let $K(t) = (A\bar{A}_t^{-1})e^{-\frac{t}{2}}$, then $\bar{q}(t,x) = q(t, K(t)^{-1}\sqrt{C}x) + f(t)$ satisfies,

$$\begin{cases} \partial_t \bar{q} - \frac{1}{2}\nabla \cdot K(t)^2\nabla\bar{q} + \frac{1}{2}|K(t)\nabla\bar{q}|^2 = 0, \\ \bar{q}(0,x) = \bar{h}(x) := -h(\sqrt{C}x). \end{cases} \tag{9}$$

It is known that the semi-concavity is preserved (Mooney et al., 2025), while the semi-convexity is not automatically propagated, which is the main difficulty in the analysis of diffusion models. In this work, we further define $\bar{p}(t,x) := e^{-\bar{q}(t,x)}$ that satisfies

$$\begin{cases} \partial_t \bar{p} - \frac{1}{2}\nabla \cdot K(t)^2\nabla\bar{p} = 0, \quad \text{on } (0,\infty) \times \mathbb{R}^n, \\ \bar{p}(0,x) = e^{-\bar{h}(x)}. \end{cases} \tag{10}$$

(10) is a heat equation and admits the following solution from the heat kernel,

$$\bar{p}(t,x) = \frac{1}{(2\pi)^{\frac{n}{2}}}\int_{\mathbb{R}^n}\frac{1}{\sqrt{\det B(t)}}\exp\left(\frac{-|x-y|_{B(t)}^2}{2}\right)\exp\left(-\bar{h}(y)\right)dy, \tag{11}$$

where $B(t) := \int_0^t K(s)^2 ds = (e^{\frac{t}{2}} - e^{-\frac{t}{2}})K(t)$. It is worth mentioning that the kernel representation (11) leads directly to uniform-in-time bounds on $\nabla\bar{q}$ and $\nabla^2\bar{q}$ (Theorem 3.1). These bounds yield the exponentially decaying Lipschitz constant of the modified score (Corollary 3.2), which is central to our Wasserstein convergence guarantees.

## 3  Results

In this section, we list the theoretical results and discussions. The detailed proofs are provided in Appendix A. Section 3.1 is devoted to the Lipschitz bound of score with Gaussian tail assumption, which is based on a heat kernel estimation of (11). Section 3.2 lists the fundamental convergence result in the Wasserstein metric. Sections 3.3 and 3.4 present applications of our convergence result to the bounded-support assumption and Bayesian inverse problems, respectively.

### 3.1 Lipschitz Bound of Score Function

The foundation of our analysis is to provide a spatially global regularity estimate of $p_t$ as the solution of an elliptic equation. These bounds control how sharply the true score can vary in space and directly impact the stability and error propagation of the learned score in the generative process. Before obtaining such bounds, we need the following assumption to derive reasonable point-wise estimates of the gradients of the score function.

**Assumption 2** (Gaussian tail). *The target density $p_0 \in C^2(\mathbb{R}^d)$ and admits the tail decomposition,*

$$p_0(x) = \exp\Big(-\frac{|x|_A^2}{2}\Big)\exp\big(h(x)\big), \tag{12}$$

*where*

    (i) *$A$ is a symmetric, positive-definite matrix which can be simultaneously diagonalized with $C$ and satisfies $\|AC^{-1}\| < \infty$ and $\|CA^{-1}\| < \infty$,*
    (ii) *The remainder term $h$ satisfies $|\sqrt{C}\nabla h|_\infty < \infty$ and $\|C\nabla^2 h\|_\infty < \infty$.*

*Moreover, all the constants above are dimension-independent.*

Similar decompositions of $p_0$ as (12) appeared in Assumption 1 in (Cole & Lu, 2024) and Theorem 13 in (Pidstrigach et al., 2024) to ensure the generalization and well-posedness of the sampling process. To analyze the convergence of the discrete sampling process and its dimensional dependence, our Assumption 2 is more restrictive, as well as comparing with the conditions used in log-concavity and heat-flow regularity theory. Its advantage lies in yielding exponentially decaying Lipschitz/Hessian bounds for the modified score, which do not follow from classical Lipschitz or weak log-concavity assumptions. More precisely, the $L$-Lipschitz condition on $\nabla \log p_0$ does not ensure a Lipschitz bound for $\nabla \log p_t$ (see Example 3.4 in (Mooney et al., 2025)), and the combination of weak log-concavity with one-sided Lipschitz-ness (Assumption H1) in (Silveri & Ocello, 2025) only guarantees an $\mathcal{O}(1)$ Hessian bound of $\log p_t$ over time. While being slightly more restrictive, the Assumption 2 yields an $\mathcal{O}(e^{-t})$ Hessian bound for modified score function (Corollary 3.2), which thus leads to improved complexity bounds (see Remark 3.6). Moreover, the Gaussian tail assumption does not require the target distribution to be log-concave, even when it is far from the origin.

To demonstrate the *practicality* of the Gaussian tail assumption, we provide two examples. First, in Theorem 3.8, we show that for any bounded-support target distribution, the density of the forward process at the early stopping time $\delta$ satisfies the assumption with $A = (1 - \exp(-\delta))I_d$. Second, the Gaussian tail assumption also holds for certain posterior distributions arising in Bayesian inverse problems; see Theorem 3.13.

Under the Gaussian tail Assumption 2, we establish bounds on the Hessian and gradient of the transformed potential $\bar{q}(t, x)$ via the heat kernel estimation.

**Theorem 3.1.** *Under Assumption 2, the function $\bar{q}(t, x)$ in (9) satisfies, $\forall t \geq 0$,*

$$|\nabla \bar{q}(t, \cdot)|_\infty \leq |\nabla \bar{q}(0, \cdot)|_\infty, \quad \|\nabla^2 \bar{q}(t, \cdot)\|_\infty \leq \|\nabla^2 \bar{q}(0, \cdot)\|_\infty + |\nabla \bar{q}(0, \cdot)|_\infty^2. \tag{13}$$

Proof see Appendix A.1.

Theorem 3.1 provides uniform in time estimates for the transformed variables in (9). By reversing the change of variables $(t, x) \to (t, (\bar{A}_t A^{-1})e^{t/2}x)$ used to go from (8) to (9), these uniform bounds translate back to the original coordinates, resulting in the exponential decay estimates for the modified score function stated in the following corollary.

**Corollary 3.2.** *Define the modified score function as*

$$\tilde{s}(t, x) := C\nabla \log p_t(x) + C\bar{A}_t^{-1}x. \tag{14}$$

*Suppose that Assumption 2 holds. Then, $\forall t \geq 0$, the following estimates hold:*

$$\|\nabla \tilde{s}(t, \cdot)\|_\infty \leq e^{-t}L_0, \quad |\tilde{s}(t, \cdot)|_\infty \leq e^{-t/2}L_1, \tag{15}$$

*where*

$$
\begin{cases}
K = \max\{1, \|AC^{-1}\|\}, \\
L_0 = K^2\big(\|C\nabla^2 h\|_\infty + |\sqrt{C}\nabla h|_\infty^2\big), \\
L_1 = K\|C\|^{1/2}|\sqrt{C}\nabla h|_\infty.
\end{cases}
$$

Proof can be found in Appendix A.2. This result provides a formal justification for a well-known empirical feature (Song et al., 2021): *the score function becomes smoother at higher noise levels.* This smoothing is crucial because (i) it enables stable training at early timesteps, and (ii) it ensures that discretization error does not explode when simulating the backward process.

Note that $K, L_0$ and $L_1$ are bounded dimension-free constants. For later discussion, we also denote,

$$
L_2 := \max\{\|I - CA^{-1}\|, \|AC^{-1} - I\|\}.
$$

*Remark* 3.3. Corollary 3.2 extends to more general linear forward SDEs of the form

$$
dX_t = -f(t)\, X_t\, dt + g(t)\, \sqrt{C}\, dB_t,
$$

where $f, g : [0, \infty) \to (0, \infty)$ are continuous functions and the solution admits the explicit representation

$$
X_t = \alpha(t)X_0 + \beta(t)Z_t, \qquad Z_t \sim \mathcal{N}(0, C),
$$

with $\alpha(t) = e^{-\int_0^t f(s)\, ds}$, and $\beta(t)^2 = \int_0^t e^{-2\int_s^t f(v)\, dv} g(s)^2\, ds$.

Assuming the same Gaussian tail condition on the initial distribution as in Assumption 2, define

$$
\bar{A}_t := \alpha(t)^2 A + \beta(t)^2 C, \qquad \tilde{s}(t, x) := C\nabla \log p_t(x) + C\bar{A}_t^{-1}x.
$$

Then the modified score $\tilde{s}(t, \cdot)$ admits uniform-in-space bounds

$$
\|\nabla \tilde{s}(t, \cdot)\|_\infty \;\leq\; \alpha(t)^2 K^2\Big(\|C\nabla^2 h\|_\infty + |\sqrt{C}\nabla h|_\infty^2\Big), \qquad |\tilde{s}(t, \cdot)|_\infty \;\leq\; \alpha(t)K\,|\sqrt{C}\nabla h|_\infty,
$$

where $K = \max\{1, \|AC^{-1}\|\}$. The OU case corresponds to the special choice $f(t) \equiv \frac{1}{2}$ and $g(t) \equiv 1$.

## 3.2 Main Convergence theories

In the preceding section, we established the crucial analytical properties of the modified score function $\tilde{s}(t, x)$ that drives the reverse diffusion process, namely the *exponential decay of its Lipschitz constant*. This section utilizes these theoretical foundations to provide rigorous, quantifiable bounds on the convergence of our discrete sampling scheme (6) under the Wasserstein-2-distance,

$$
\mathcal{W}_2(\mu, \nu) := \left(\inf_\pi \int |x - y|^2 d\pi(x, y)\right)^{1/2},
$$

where $\pi$ runs over all measures which have marginals $\mu$ and $\nu$. Our objective is to clearly dissect the sources of the total sampling error and demonstrate how this error decomposes and evolves with respect to the diffusion time $T$, the step size $\tau$, and the score network approximation accuracy $\epsilon$. To achieve this, we first introduce the following assumptions concerning the learned score function $s_\theta$. Assumption 3 quantifies how accurately the learned score approximates the true score at discretization points. Assumption 4 ensures a Lipschitz-type regularity for the learned score, consistent with the theoretically guaranteed regularity of the true score.

**Assumption 3.** *$s_\theta$ is an $\epsilon$-accurate approximation to $s$ on average over the discretization points, i.e.,*

$$
\frac{1}{T}\sum_{k=0}^{N-1}(t_{k+1} - t_k)\mathbb{E}\Big|s(T - t_k, \overrightarrow{X}_{T-t_k}) - s_\theta(T - t_k, \overrightarrow{X}_{T-t_k})\Big|^2 \leq \epsilon^2. \tag{16}
$$

We would also like to point out that, due to the limited access to the score approximation error, when deriving the complexity bounds, we always assume a sufficiently accurate score approximation.

For the subsequent analysis, we will work with the modified score $\tilde{s}(t,x) := s(t,x) + C\bar{A}_t^{-1}x$ and its approximation $\tilde{s}_\theta(t,x) := s_\theta(t,x) + C\bar{A}_t^{-1}x$. Note that adding the same deterministic correction $C\bar{A}_t^{-1}x$ to both terms does not change the approximation error. Hence Assumption 3 is equivalently stated in terms of $(\tilde{s}, \tilde{s}_\theta)$; the network being trained is still $s_\theta$, while $\tilde{s}_\theta$ is introduced only for stability analysis.

Assumption 3 evaluates the score approximation error along the true reverse-time process. This choice is aligned with the learning objective in the denoising score matching (Ho et al., 2020), where expectations are taken with respect to the true diffusion process $\overrightarrow{X}$ rather than the empirical sampling trajectory $\overleftarrow{Y}$. Alternative choices, which evaluate the score approximation error along $\overleftarrow{Y}$, have been used in the literature (Gao et al., 2025; Yu & Yu, 2025; Silveri & Ocello, 2025) and simplify the control of error propagation, but they measure the approximation error on a path that itself depends on the learned score network $s_\theta$.

Under the present formulation, the propagation of errors between successive time steps is governed by the discrete flow induced by the learned score. As a consequence, in addition to Assumption 3, some regularity of the score network is required to ensure stability of the propagation. This motivates the following Assumption 4, which postulates that the learned score satisfies a time-decaying Lipschitz bound consistent with that of the true score established in Corollary 3.2. We note that similar considerations are also discussed in equation (3.3) of Gao et al. (2025) and in Section 4 of Beyler & Bach (2025).

**Assumption 4.** *We assume that $\tilde{s}_\theta(t,x)$ satisfies the same time-decaying Lipschitz bound as $\tilde{s}(t,x)$ in Corollary 3.2 at the time discretization points, i.e.*

$$\forall k \in \{0, 1, \dots, N-1\}, \quad \|\nabla \tilde{s}_\theta(T - t_k, \cdot)\|_\infty \le L_0 e^{-T+t_k}. \tag{17}$$

The Assumption 4 provides a sufficient regularity assumption to establish later convergence results and it is consistent with the theoretical estimates in Corollary 3.2. Without prior knowledge of $A$ and $C$, (17) may be difficult to validate, and can be relaxed to the following condition,

**Assumption 4′.** *There exists some constants $b > 0$ and $B > 0$ independent of $T$, such that*

$$\sup_k \|I + \nabla s_\theta(T - t_k, \cdot)\|_\infty \le b \quad and \quad \sum_{k=0}^{N-1} (t_{k+1} - t_k)\|I + \nabla s_\theta(T - t_k, \cdot)\|_\infty \le B. \tag{18}$$

We note that the relaxed condition, formulated as a time-integrated stability bound, captures the actual requirement used in the convergence analysis. Moreover, it is closely aligned with practice. For example, Lipschitz-controlled architectures such as spectral-normalized networks (Miyato et al., 2018) provide an explicit control of Lipschitz bounds on $x + s_\theta(t,x)$, which, possibly combined with time-dependent scaling in terms of $e^{-t}$, yield the required stability control. In the later presentation of the main results, we will take Assumption 4. In Appendix A.4, we provide the adaptation of the proof to Assumption 4′.

Now we present the main theorem in this part.

**Theorem 3.4.** *Suppose Assumptions 1, 2, 3, and 4 hold, with the step size $\tau := \sup_k\{t_{k+1} - t_k\} \le 1$. Then sampling via scheme* (6) *yields*

$$\mathcal{W}_2^2(\overrightarrow{P}_0, \overleftarrow{Q}_T) \le K_1\left(e^{-2T}\left(M_2 + \mathrm{Tr}(C)\right) + \epsilon^2 T + (L_0 + L_2)^2\mathrm{Tr}(C)\tau^2\right), \tag{19}$$

*where $K_1 := \exp\left(4 + (1+e)(2(L_0 + L_2) + (L_0 + L_2)^2 e\tau)\right)$ is dimension-free.*

Proof see Appendix A.3.

This theorem decomposes the total sampling error into several contributions. The prefactor $K_1$ accounts for the accumulation of errors through the reverse-time process. It arises from a discrete-time propagation analysis based on a Grönwall-type argument and depends on the Lipschitz regularity of the modified score, which is ensured Assumption 4. The remaining terms correspond to additive error contributions: (i) $e^{-2T}(M_2 + \mathrm{Tr}(C))$

is the error due to incomplete denoising, (ii) $\epsilon^2 T$ arises from the score approximation error and is governed by Assumption 3. (iii) $(L_0 + L_2)^2 \mathrm{Tr}(C)\tau^2$ is the discretization error, whose magnitude is amplified by the Lipschitz constant of the reverse-time drift guaranteed under the Gaussian tail Assumption 2. Practitioners may choose mixing time $T$, network size, and timestep schedules $\tau$ accordingly. We also note that when applying Theorem 3.4 to the convergence with respect to $\tau \to 0$, the constant $\log K_1$ may be made linear in $L_0 + L_2$ with $\tau$ sufficiently small. In Section 3.3, we will consider such a growth regime for the targets with bounded supports.

As a direct consequence of Theorem 3.4, we have the following complexity bound for Gaussian tail targets.

**Corollary 3.5.** *Under the same assumptions as Theorem 3.4, the following complexity bound guarantees that the distribution $\overleftarrow{Q}_T$ satisfies $\mathcal{W}_2(\overrightarrow{P}_0, \overleftarrow{Q}_T) = \mathcal{O}(\epsilon_0)$:*

$$T = \mathcal{O}\left( \log \frac{M_2 + \mathrm{Tr}(C)}{\epsilon_0^2} \right), \quad N = \mathcal{O}(\frac{T}{\tau}) = \mathcal{O}\left( \frac{T}{\epsilon_0} \sqrt{\mathrm{Tr}(C)} \right). \tag{20}$$

*Remark* 3.6. When assuming the second moment of the target distribution $M_2$ and the trace of the diffusion covariance $\mathrm{Tr}(C)$ scale linearly with space dimension $d$, the bound in Theorem 3.4 corresponds to an $\mathcal{O}(\tau)$ convergence rate with $\mathcal{O}(\sqrt{d})$ dependence in $\mathcal{W}_2$ distance, and the complexity bound in Corollary 3.5 is $\mathcal{O}(\sqrt{d})$ with a logarithmic constant. In (Gao et al., 2025), they obtained an $\mathcal{O}(d)$ complexity under an assumption similar to our Gaussian-tail assumption (they further assume $h$ is concave), and their Proposition 6 shows that the $\mathcal{O}(\sqrt{d})$ bound is optimal when the target is the standard Gaussian. Notably, in the line of pursuing complexity bounds under more general assumptions (than log-concaveness), a very recent work (Silveri & Ocello, 2025) obtained an $\mathcal{O}(d^2)$ bound[2] through the weakly log-concave profile propagation framework established in (Saremi et al., 2024; Conforti, 2024; Conforti et al., 2025). In Table 1, we provide a non-exhaustive list of complexity bounds for comparison.

When fixing the second moment $M_2$ and trace of covariance matrix $C$ of base distribution, the complexity bound in Corollary 3.5 does not depend on the dimension and hence can be generalized to some infinite-dimensional generative models; see Theorem B.3 and related discussion in Appendix B.

*Remark* 3.7. In Corollary 3.5, we implicitly operate in a regime where the parameters $L_0$ and $L_2$ are moderate, so that the prefactor $K_1$, which depends exponentially on $L_0 + L_2$, does not dominate the error bound. This situation corresponds to typical smooth-data settings, in which $L_0$ and $L_2$ are determined by derivatives of the log-density and by the consistency between the base distribution and the target covariance structure.

By contrast, in regimes involving bounded-support targets or early stopping, the quantities $L_0$ and $L_2$ may deteriorate as the support radius increases or the stopping parameter decreases. In such cases, the exponential dependence of $K_1$ is unavoidable and reflects a genuine loss of stability of the reverse-time process. We refer to Section 3.3 for a more detailed discussion, where we analyze targets with bounded support and identify parameter regimes in which the bounds remain informative.

### 3.3 Convergence for bounded-support target

This subsection specializes the convergence theory of Theorem 3.4 to an important and practically relevant setting: when the target distribution $p_0$ is supported on a bounded set. Such distributions arise naturally in many practical applications of generative models and can be derived from the manifold hypothesis (Bengio et al., 2013). We make this assumption explicit as follows.

**Assumption 5.** *The target distribution $\overrightarrow{P}_0$ is supported on a bounded set, i.e., there exists $R < \infty$ such that*

$$\mathrm{Supp}\,\overrightarrow{P}_0 \subset B(0, R).$$

While Assumption 5 alone does not impose any smoothness or tail behavior on $p_0$, a key phenomenon is that the forward Ornstein–Uhlenbeck diffusion instantaneously regularizes such distributions. In particular, for every $t > 0$, the density $p_t$ acquires Gaussian-type tails and thus satisfies the Gaussian tail condition of

---

[2]In arriving at this, we also assume the second moment scales linearly with dimension in their Theorem 3.5.

Assumption 2. To quantify this regularizing effect, we first record a general result on Gaussian convolution of a bounded-support distribution.

**Theorem 3.8.** *Let $Q_\sigma = \mathcal{N}(0, \sigma^2 I_d) * Q_0$, where $Q_0$ follows Assumption 5. Then $Q_\sigma$ has smooth density $q_\sigma$ and we define*

$$g(x) := \log q_\sigma(x) + \frac{|x|^2}{2\sigma^2}.$$

*It follows that,*

$$|\nabla g|_\infty \leq \frac{R}{\sigma^2}, \quad \|\nabla^2 g\|_\infty \leq \frac{2R^2}{\sigma^4}. \tag{21}$$

Proof see Appendix A.5.

Similar estimates in Theorem 3.8 can be found in (De Bortoli, 2022; Mooney et al., 2025). In the current form (21), we extract the spatial growing part to ensure uniform boundedness in space to be consistent with Assumption 2.

Applying Theorem 3.8 to the forward OU-process (1) with covariance $C = I_d$ and stopping time $\delta$, we obtain that the marginal $\overrightarrow{P}_\delta$ has smooth density $p_\delta$ and satisfies,

$$p_\delta(x) = \exp\left(-\frac{|x|^2}{2(1 - e^{-\delta})}\right) \exp\left(h(x)\right),$$

and

$$|\nabla h|_\infty \leq \frac{e^{-\frac{\delta}{2}} R}{1 - e^{-\delta}}, \qquad \|\nabla^2 h\|_\infty \leq 2\frac{e^{-\delta} R^2}{(1 - e^{-\delta})^2}.$$

Hence $p_\delta$ satisfies Assumption 2, and the score Lipschitz bounds from Corollary 3.2 apply directly.

**Corollary 3.9.** *Suppose that $\overrightarrow{P}_0$ follows the Assumption 5. Let $C = I_d$ and $A = (1 - e^{-\delta})I_d$. Then $\bar{A}_t = (1 - e^{-t})I_d$ and for all $t \geq \delta$, we have,*

$$\|\nabla^2 \log p_t(x) + \bar{A}_t^{-1}\|_\infty \leq 3\left(\frac{R}{1 - e^{-\delta}}\right)^2 e^{-t}, \quad |\nabla \log p_t(x) + \bar{A}_t^{-1}x|_\infty \leq \frac{R}{1 - e^{-\delta}} e^{-\frac{t}{2}}.$$

Proof see Appendix A.6. As shown in Corollary 3.9, even though the score of an early-stopped bounded-support distribution is well defined for any positive stopping time, its global regularity deteriorates as the stopping time tends to 0. By contrast, the score of a full-support distribution with Gaussian tails admits stable global regularity properties. This behavior is closely related to the score-regularity phenomena discussed in Stéphanovitch (2025).

Using the Lipschitz bound in Corollary 3.9, we obtain a Wasserstein-2 distance bound analogous to Theorem 3.4.

**Theorem 3.10.** *Suppose that $\overrightarrow{P}_0$ follows the Assumption 5, the early stopping time $\delta \leq 1$, and the Assumption 3 holds. Then, sampling via scheme (6) with step size $\tau$ satisfying*

$$\tau \leq \min\{1, \left(3e\left(\frac{R}{1 - e^{-\delta}}\right)^2 + \frac{e}{1 - e^{-\delta}}\right)^{-1}\},$$

*we obtain the following convergence bound,*

$$\mathcal{W}_2^2(\overrightarrow{P}_\delta, \overleftarrow{Q}_{T-\delta}) \leq \exp\left(4 + 3(1 + e)K_2\right)\left(e^{-2T+\delta}\left(R^2 + d\right) + \epsilon^2 T + K_2^2 d\tau^2\right), \tag{22}$$

*where $K_2 = 3\frac{R^2}{(1-e^{-\delta})^2} + \frac{1}{1-e^{-\delta}}$.*

Proof see Appendix A.7. With the result in Theorem 3.10, we can get the complexity bound with early stopping setting under the bounded-support assumption by direct computation.

**Corollary 3.11.** *Under the same assumption as Theorem 3.10, achieving a distribution $\overleftarrow{Q}_{T-\delta}$ such that $\mathcal{W}_2(\overrightarrow{P}_\delta, \overleftarrow{Q}_{T-\delta}) = \mathcal{O}(\epsilon_0)$ requires:*

$$T = \mathcal{O}\Big(\frac{R^2}{\delta^2} + \log\frac{R^2 + d}{\epsilon_0^2}\Big), \quad N = \frac{T - \delta}{\tau} = \mathcal{O}\left(\frac{\sqrt{d}T}{\epsilon_0}\exp\left(\frac{1 + e}{1 - e^{-\delta}} + \frac{3(1 + e)R^2}{(1 - e^{-\delta})^2}\right)\right).$$

*When fixing the early stopping time $\delta$ and support radius $R$,*

$$N = \mathcal{O}\Big(\frac{\sqrt{d}}{\epsilon_0}\log\frac{R^2 + d}{\epsilon_0^2}\Big). \tag{23}$$

Noticing that, $\mathcal{W}_2(\overrightarrow{P}_\delta, \overrightarrow{P}_0) \leq \sqrt{\mathbb{E}|\overrightarrow{X}_\delta - \overrightarrow{X}_0|^2} \leq 2\sqrt{M_0\delta}$, we have the following complexity bound with respect to $\overrightarrow{P}_0$.

**Corollary 3.12.** *Under the same assumption as Theorem 3.10, reaching a distribution $\overleftarrow{Q}_{T-\delta}$ such that $\mathcal{W}_2(\overrightarrow{P}_0, \overleftarrow{Q}_{T-\delta}) = \mathcal{O}(\epsilon_0)$ requires,*

$$\delta = \mathcal{O}\Big(\frac{\epsilon_0^2}{M_0}\Big), \quad \log N = \mathcal{O}\Big(\frac{R^2}{\delta^2}\Big) = \mathcal{O}\Big(\frac{R^2 M_0^2}{\epsilon_0^4}\Big). \tag{24}$$

The logarithmic complexity $\log N = \mathcal{O}\big(\frac{R^2}{\delta^2}\big)$ stems from the exponential dependence of the prefactor $K_2(R, \delta)$ on $\frac{R^2}{(1 - e^{-\delta})^2}$, which can be traced to two sources. First, the Hessian bound for the early-stopped log-density already scales as $\frac{R^2}{(1 - e^{-\delta})^2}$ (Corollary 3.9); to the best of our knowledge, this $(R, \delta)$-dependence cannot be improved without additional geometric assumptions on $\text{Supp}(\overrightarrow{P}_0)$, see, e.g., (Mooney et al., 2025). Second, discretization error accumulates through the Lipschitz constant of the modified score in the reverse-time process: since this Lipschitz constant is controlled by the Hessian bound, the discrete-time stability analysis (via a Grönwall-type argument) induces an exponential amplification. Such exponential dependence is classical in Wasserstein-based error analyses and in Lipschitz change-of-variable estimates for deterministic or stochastic flows (De Bortoli, 2022; Beyler & Bach, 2025; Brigati & Pedrotti, 2025). In the absence of additional global contractivity assumptions (e.g., strong convexity), this type of dependence is generally unavoidable.

By contrast, KL/TV-based analyses that exploit Girsanov-type arguments may yield only polynomial dependence on $\frac{R}{\delta}$ when converted to $W_2$ bounds (Chen et al., 2023b; Holden et al., 2023). However, these guarantees typically apply to *projected* distributions (discarding mass outside $B_0(R)$ to pass from TV/KL to Wasserstein), whereas the present work controls the *unprojected* law of the reverse-time process. In this setting, we are not aware of a simple way to control the projection error, and results for projected and unprojected distributions are therefore not directly comparable. For completeness, Appendix D discusses how Corollary 3.2 can be used to derive an upper bound with a similar $(R, d)$-dependence as (Chen et al., 2023b), and Table 2 summarizes a broader comparison under bounded-support assumptions.

### 3.4 Convergence in the Bayesian Inverse problems

Another potential application of generative models is to generate the posterior distribution in Bayesian inverse problems. See (Stuart, 2010) for a detailed review. Here, we restrict our theories to the following type of applicative scenario, where we consider a non-linear observation $G \in C_2^b(\mathbb{R}^d, \mathbb{R}^m)$ from the state space $\mathbb{R}^d$ and its observation $y \in \mathbb{R}^m$. We further assume that the prior of the state and the observation noise distribution follow Gaussian-type distributions with $C$ and $\Sigma$ denoting the covariance matrices, respectively. Then the posterior of the state, also the target distribution of the generative model, follows,

$$p_0(x) = D_0 \exp\Big(-\frac{|x|_C^2}{2}\Big)\exp\Big(-\frac{|G(x) - y|_\Sigma^2}{2}\Big), \tag{25}$$

where $D_0$ is some normalizing constant.

To construct a generative model to sample the posterior, we take the covariance matrix of the base distribution in (1) identical to the prior covariance matrix $C$ in (25). A conditioned score (referred to as the conditional de-noising estimator in the literature (Batzolis et al., 2021)) is trained with the following loss,

$$\mathbb{E}_{p_t(x;y)}|s_\theta(t, x; y) - C\nabla_x \log p_t(x; y)|^2,$$

where $p_t$ is then the joint distribution of $(X_t, Y)$ in which $Y$ follows $G(X_0) + \mathcal{N}(0, \Sigma)$. For the generation process of the posterior distribution with observation $y$, we assume the estimated score $s_\theta(t, x; y)$ satisfies Assumption 3 and 4. Then, we have the following theorem.

**Theorem 3.13.** *Suppose that the target distribution admits the density in* (25) *and Assumption 1 holds. Then, sampling via scheme* (6) *with step size $\tau$ satisfying*

$$\tau \leq \min\{1, \|\Sigma\|\|C\|^{-1}\Big(\big(\|\nabla^2 G\|_\infty(|G|_\infty + |y|) + \|\nabla G\|_\infty^2\big) + \|\Sigma\|^{-1}\|\nabla G\|_\infty^2\big(|G|_\infty + |y|\big)^2\Big)^{-1}\},$$

*we obtain the following convergence bound,*

$$\mathcal{W}_2^2\big(\overrightarrow{P}_0(\cdot; y), \overleftarrow{Q}_T(\cdot; y)\big) \leq K_3\Big(e^{-2T}\big(M_2 + \mathrm{Tr}(C)\big) + \epsilon^2 T + k_3^2\mathrm{Tr}(C)\tau^2\Big), \tag{26}$$

*where $K_3$ and $k_3$ are dimensionless constants determined by $(\|C\|, \|\Sigma\|, G, y)$, the explicit expression is provided in Table 3.*

Proof see Appendix A.8.

This provides a theoretical grounding for the growing empirical use of diffusion models as posterior samplers (e.g. conditional diffusion models), justifying that with sufficiently accurate score learning, the generated posterior approximations are provably close to the true Bayesian posterior in the Wasserstein-2-distance.

## 4 Conclusion

In this work, we presented a unified and generalizable theoretical framework for analyzing the convergence of score-based generative diffusion models (SGMs), specifically focusing on bounding the Wasserstein-2 distance between the target and generated distributions. Our analysis hinges on the introduction of the *Gaussian-tail assumption* (Assumption 2), a more general condition that accommodates practical scenarios such as distributions with bounded support and those arising from early stopping techniques. A core technical contribution is establishing the *exponential decay over time* of the Lipschitz constant for the modified score function (Corollary 3.2), derived via a dimension-independent heat kernel estimate. This enabled us to prove a rigorous Wasserstein-2 distance bound (Theorem 3.4) that only depends on the second moment of the base distribution, thereby naturally extending the analysis to the infinite-dimensional setting. Furthermore, for the standard finite-dimensional case, our analysis demonstrates an optimal iterative complexity rate of $\mathcal{O}(\sqrt{d})$ (Corollary 3.5), matching the current state-of-the-art. Overall, our work provides stringent theoretical guarantees, affirming the convergence and optimal dimension-dependence of the discrete approximation scheme for SDE-based score models under a more general class of target distributions, judged by the Wasserstein-2 metric.

Moving forward, a compelling direction for future research involves investigating the higher-order regularity of the score function. While our current analysis provides regularity estimates up to the second derivative—corresponding to the second-order boundedness of the Gaussian-tail's remainder term—we believe that imposing higher-order regularity on this remainder term (which is feasible in practice via bounded support combined with early stopping) could yield analogous higher-order regularity estimates for the score function itself. Such advanced regularity understanding is beneficial for guiding neural network training objectives and enabling the discovery of higher-order numerical SDE schemes, thereby significantly improving the convergence rate and accuracy of the discrete numerical scheme relative to the true backward diffusion process.

## Broader Impact

The paper provides insights into the regularity and convergence of one of the state-of-the-art generative models, which the GenAI community may benefit from. There are many potential societal consequences of our work, none of which we feel must be specifically highlighted here.

## Acknowledgment

The research of both authors is partially supported by NTU-SUG and MOE AcRF Tier 1 RG17/24.

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

## Appendix

The appendix consists of three parts. In Section A, we present all the detailed proofs. In Section B, we discuss generalizing our theories to infinite dimensions. In Section C, we provide the tables mentioned in the context.

## A  Proofs of Theorems

Here we present detailed proofs.

### A.1  Proof of Theorem 3.1 (heat kernel estimation)

*Proof.* Consider the solution of (10) given by (11),

$$\bar{p}(t,x) = \frac{1}{(2\pi)^{\frac{n}{2}}} \int_{\mathbb{R}^n} \frac{1}{\sqrt{\det B(t)}} \exp\Big(\frac{-|x-y|^2_{B(t)}}{2}\Big) \exp\big(-\bar{h}(y)\big) dy, \quad (t,x) \in (0,\infty) \times \mathbb{R}^n.$$

So $\bar{q}(t,x) = -\log \bar{p}(t,x)$ satisfies,

$$
\begin{aligned}
\nabla_x \bar{q}(t,x) &= -\frac{\nabla_x \bar{p}(t,x)}{\bar{p}(t,x)} = -\frac{\int_{\mathbb{R}^n} \big(\nabla_x \exp(\frac{-|x-y|^2_{B(t)}}{2})\big) \exp\big(-\bar{h}(y)\big) dy}{\int_{\mathbb{R}^n} \exp(\frac{-|x-y|^2_{B(t)}}{2}) \exp\big(-\bar{h}(y)\big) dy} \\
&= \frac{\int_{\mathbb{R}^n} \big(\nabla_y \exp(\frac{-|x-y|^2_{B(t)}}{2})\big) \exp\big(-\bar{h}(y)\big) dy}{\int_{\mathbb{R}^n} \exp(\frac{-|x-y|^2_{B(t)}}{2}) \exp\big(-\bar{h}(y)\big) dy} \\
&= \frac{\int_{\mathbb{R}^n} \big(\nabla_y \bar{h}(y)\big) \exp(\frac{-|x-y|^2_{B(t)}}{2}) \exp\big(-\bar{h}(y)\big) dy}{\int_{\mathbb{R}^n} \exp(\frac{-|x-y|^2_{B(t)}}{2}) \exp\big(-\bar{h}(y)\big) dy}.
\end{aligned}
\tag{27}
$$

Here, the third line is derived from the integration by parts formula.

Since $\exp(\frac{-|x-y|^2_{B(t)}}{2}) \exp\big(-\bar{h}(y)\big) \geq 0$, taking absolute value we get $|\nabla_x \bar{q}(t,x)| \leq |\nabla \bar{h}|_\infty$, thus,

$$|\nabla \bar{q}(t,\cdot)|_\infty \leq |\nabla \bar{h}|_\infty = |\nabla \bar{q}(0,\cdot)|_\infty.$$

For any unit direction $z$, we denote $\nabla_z$ as taking derivative along that direction and $\nabla_z$ as taking derivative twice along that direction, i.e., for a given $C^2$ function $f$,

$$\nabla_z f(x) = \langle z, \nabla f(x)\rangle, \quad \nabla_z^2 f(x) = \langle z, \nabla^2 f(x) z\rangle,$$

where $\nabla f(x)$ is the gradient of $f$ and $\nabla^2 f(x)$ is the Hessian matrix of $f$ at $x$. Using the same method as above, we get,

$$
\begin{aligned}
\nabla_z^2 \bar{q}(t,x) &= \frac{\int_{\mathbb{R}^n} \big(\nabla_z^2 \bar{h} - (\nabla_z \bar{h})^2\big)(y) \exp(\frac{-|x-y|^2_{B(t)}}{2}) \exp\big(-\bar{h}(y)\big) dy}{\int_{\mathbb{R}^n} \exp(\frac{-|x-y|^2_{B(t)}}{2}) \exp\big(-\bar{h}(y)\big) dy} \\
&+ \Big(\frac{\int_{\mathbb{R}^n} \big(\nabla_z \bar{h}(y)\big) \exp(\frac{-|x-y|^2_{B(t)}}{2}) \exp\big(-\bar{h}(y)\big) dy}{\int_{\mathbb{R}^n} \exp(\frac{-|x-y|^2_{B(t)}}{2}) \exp\big(-\bar{h}(y)\big) dy}\Big)^2.
\end{aligned}
\tag{28}
$$

Taking absolute value again, we get,

$$|\nabla_z^2 \bar{q}(t,x)| \leq |\nabla_z^2 \bar{h}|_\infty + |\nabla_z \bar{h}|_\infty^2 \leq \|\nabla^2 \bar{h}\|_\infty + |\nabla \bar{h}|_\infty^2, \quad \forall x \in \mathbb{R}^n,$$

thus,

$$\|\nabla^2 \bar{q}(t,x)\| = \sup_{|z|=1} |\nabla_z^2 \bar{q}(t,x)| \leq \|\nabla^2 \bar{h}\|_\infty + |\nabla \bar{h}|_\infty^2, \quad \forall x \in \mathbb{R}^n.$$

Since the bound for $\|\nabla^2 \bar{q}(t,x)\|$ does not rely on $x$, we get,

$$\|\nabla^2 \bar{q}(t,\cdot)\|_\infty \leq \|\nabla^2 \bar{h}\|_\infty + |\nabla \bar{h}|_\infty^2 = \|\nabla^2 \bar{q}(0,\cdot)\|_\infty + |\nabla \bar{q}(0,\cdot)|_\infty^2. \tag{29}$$

$\square$

The above analysis is developed to facilitate the case $C \neq I_d$; other standard PDE approaches, like the classical Bernstein method, can also be applied for the isotropic case, i.e., $C = I_d$.

*Remark* A.1. The representations (27) and (28) can also be obtained from the Gaussian convolution formula for $p_t$ by factoring out the Gaussian part and applying integration by parts to the remainder term. While this direct computation is feasible, it does not make the underlying mechanism as transparent. The vHJ formulation reveals the PDE structure behind score regularity and highlights the key difficulty: semi-concavity can be propagated, whereas semi-convexity is not automatically propagated. On the other hand, the heat-kernel representation makes the parabolic regularization explicit, so that the desired Lipschitz and Hessian bounds follow from the well-known positivity and averaging properties of the heat kernel.

## A.2 Proof of Corollary 3.2

*Proof.* Recall that $\bar{q}(t,x) = q(t, K(t)^{-1}\sqrt{C}x) + f(t)$, so,

$$K(t)\sqrt{C^{-1}}\nabla\bar{q}(t,x) = \nabla q(t, K(t)^{-1}\sqrt{C}x),$$
$$K(t)^2 C^{-1}\nabla^2\bar{q}(t,x) = \nabla^2 q(t, K(t)^{-1}\sqrt{C}x).$$

Notice that

$$\tilde{s}(t,x) = C\nabla q(t,x), \quad \nabla\tilde{s}(t,x) = C\nabla^2 q(t,x).$$

Hence,

$$|\tilde{s}(t,\cdot)|_\infty = |C\nabla q(t,\cdot)|_\infty = |K(t)\sqrt{C}\nabla\bar{q}(t,\cdot)|_\infty \leq e^{-\frac{t}{2}}\|e^{\frac{t}{2}}K(t)\|\|\sqrt{C}\||\nabla\bar{q}(t,\cdot)|_\infty,$$
$$\|\nabla\tilde{s}(t,\cdot)\|_\infty = \|C\nabla^2 q(t,\cdot)\|_\infty = \|K(t)^2\nabla^2\bar{q}(t,\cdot)\|_\infty \leq e^{-t}\|e^{\frac{t}{2}}K(t)\|^2\|\nabla^2\bar{q}(t,\cdot)\|_\infty.$$

Define the constant,

$$K := \sup_{t\geq 0}\|e^{\frac{t}{2}}K(t)\| = \sup_{t\geq 0}\|A\bar{A}_t^{-1}\| = \max\{1, \|AC^{-1}\|\}.$$

By Theorem 3.1 and the initial value $\bar{q}(0,x) = h(\sqrt{C}x)$, we obtain,

$$|\tilde{s}(t,\cdot)| \leq e^{-\frac{t}{2}}K\|C\|^{\frac{1}{2}}|\sqrt{C}\nabla h|_\infty,$$
$$\|\nabla\tilde{s}(t,\cdot)\|_\infty \leq e^{-t}K^2\left(\|C\nabla^2 h\|_\infty + |\sqrt{C}\nabla h|_\infty^2\right). \tag{30}$$

Let,

$$L_0 := K^2\left(\|C\nabla^2 h\|_\infty + |\sqrt{C}\nabla h|_\infty^2\right),$$
$$L_1 := K\|C\|^{\frac{1}{2}}|\sqrt{C}\nabla h|_\infty,$$

then we get the results in Corollary 3.2. $\square$

## A.3 Proof of Theorem 3.4

We first derive some estimates that will be useful in the proof. Since

$$\sup_{t\geq 0}\|e^t(I - C\bar{A}_t^{-1})\| = \sup_{t\geq 0}\|(A-C)\left(Ae^{-t} + C(1-e^{-t})\right)^{-1}\|$$
$$= \max\{\|I - CA^{-1}\|, \|AC^{-1} - I\|\},$$
$$= L_2.$$

We have the following estimate,

$$\|I - C\bar{A}_t^{-1}\| \leq L_2 e^{-t}, \quad \forall t \geq 0. \tag{31}$$

Denote,

$$\hat{s}(t,x) := s(t,x) + x, \quad \hat{s}_\theta(t,x) := s_\theta(t,x) + x.$$

Recall from Corollary 3.2 that, for all $t \geq 0$,

$$\|\nabla\tilde{s}(t,\cdot)\|_\infty \leq L_0 e^{-t}, \quad |\tilde{s}(t,\cdot)|_\infty \leq L_1 e^{-\frac{t}{2}},$$

then,

$$\|\nabla\hat{s}(t,\cdot)\|_\infty = \|\nabla s(t,\cdot) + I\|_\infty \leq \|\nabla\tilde{s}(t,\cdot)\|_\infty + \|I - C\bar{A}_t^{-1}\| \leq (L_0 + L_2)e^{-t}. \tag{32}$$

Similarly, under Assumption 4, we have,

$$\forall k \in \{0,1,...,N-1\}, \quad \|\nabla\hat{s}_\theta(T - t_k, \cdot)\|_\infty = \|\nabla s_\theta(T - t_k, \cdot) + I\|_\infty \leq (L_0 + L_2)e^{-T + t_k}. \tag{33}$$

Throughout this proof, we set the step size $\tau := \sup_k(t_{k+1} - t_k) \leq 1$, so that

$$\forall r > 0, \quad \frac{e^{r\tau} - 1}{r} \leq e^r \tau. \tag{34}$$

Here, we present a lemma related to convergence to equilibrium for the OU process.

**Lemma A.2** (Theorem 23.26 (Villani et al., 2009)). *Let $V$ be $\lambda$-uniformly convex $C^2$ potential. Consider the Langevin process,*

$$dX_t = -\frac{1}{2}\nabla V(X_t)dt + \sqrt{C}dW_t,$$

*with two initial measure $\mu_0$ and $\nu_0$.*

$$\mathcal{W}_2(\mu_t, \nu_t) \leq \mathcal{W}_2(\mu_0, \nu_0)e^{-\frac{\lambda t}{2}}.$$

The convergence of OU is a direct consequence, with $\lambda = 1$. We also need the following martingale property for the score function.

**Lemma A.3.** *The quantity $e^{-\frac{t}{2}}\left(s(T - t, \overleftarrow{X}_t) + \overleftarrow{X}_t\right)$ satisfies,*

$$de^{-\frac{t}{2}}\left(s(T - t, \overleftarrow{X}_t) + \overleftarrow{X}_t\right) = e^{-\frac{t}{2}}\left(I + \nabla s(T - t, \overleftarrow{X}_t)\right)\sqrt{C}d\tilde{B}_t, \tag{35}$$

*Moreover, since (35) expresses $e^{-\frac{t}{2}}\left(s(T - t, \overleftarrow{X}_t) + \overleftarrow{X}_t\right)$ as a stochastic integral with no drift term, the process is a martingale.*

*Proof.* By Itô's formula and (4),

$$ds(T - t, \overleftarrow{X}_t) = -\partial_t s(T - t, \overleftarrow{X}_t)dt + \nabla s(T - t, \overleftarrow{X}_t)d\overleftarrow{X}_t + \frac{1}{2}C\nabla\left(\nabla \cdot s(T - t, \overleftarrow{X}_t)\right)dt$$

$$= -\partial_t s(T - t, \overleftarrow{X}_t) + \frac{1}{2}\left(\nabla s(T - t, \overleftarrow{X}_t)\right)\overleftarrow{X}_t dt + \left(\nabla s(T - t, \overleftarrow{X}_t)\right)s(T - t, \overleftarrow{X}_t)dt$$

$$+ \left(\nabla s(T - t, \overleftarrow{X}_t)\right)\sqrt{C}dB_t + \frac{1}{2}C\nabla\left(\nabla \cdot s(T - t, \overleftarrow{X}_t)\right)dt.$$

From the Fokker-Planck equation, $\partial_t p = \frac{1}{2}\left(\nabla \cdot (xp + C\nabla p)\right)$, we have,

$$
\begin{aligned}
\partial_t s =& \partial_t C \nabla \log p = C\left(\frac{\nabla \partial_t p}{p} - \frac{\partial_t p}{p^2}\nabla p\right)\\
=& \frac{1}{2}C\left(\frac{\nabla\left(\nabla \cdot (xp + C\nabla p)\right)}{p} - \frac{\nabla \cdot (xp + C\nabla p)}{p^2}\nabla p\right)\\
=& \frac{1}{2}C\left(\frac{\nabla\left(dp + x \cdot \nabla p + \nabla \cdot p\left(C\nabla \log p\right)\right)}{p} - \frac{dp + x \cdot \nabla p + \nabla \cdot (pC\nabla \log p)}{p^2}\nabla p\right)\\
=& \frac{1}{2}C\left(d\nabla \log p + \frac{\nabla\left(x \cdot p\nabla \log p + (\nabla p) \cdot C\nabla \log p + p\nabla \cdot C\nabla \log p\right)}{p}\right.\\
& \left. -d\nabla \log p - (x \cdot \nabla \log p)\nabla \log p - \frac{(\nabla p)\cdot C\nabla \log p + p\nabla \cdot C\nabla \log p}{p}\nabla \log p\right)\\
=& \frac{1}{2}C\left(\nabla \log p + (x \cdot \nabla \log p)\nabla \log p + \left(\nabla^2 \log p\right)x + \frac{\nabla\left((p\nabla \log p)\cdot C\nabla \log p\right)}{p}\right.\\
& + (\nabla \cdot C\nabla \log p)\nabla \log p + \nabla\left(\nabla \cdot C\nabla \log p\right) - (x \cdot \nabla \log p)\nabla \log p\\
& \left. - (\nabla \log p \cdot C\nabla \log p)\nabla \log p - (\nabla \cdot C\nabla \log p)\nabla \log p\right)\\
=& \frac{1}{2}C\left(\nabla \log p + \left(\nabla^2 \log p\right)x + (\nabla \log p \cdot C\nabla \log p)\nabla \log p + 2\left(\nabla^2 \log p\right)C\nabla \log p\right.\\
& \left. + \nabla\left(\nabla \cdot C\nabla \log p\right) - (\nabla \log p \cdot C\nabla \log p)\nabla \log p\right)\\
=& \frac{1}{2}C\left(\nabla \log p + \left(\nabla^2 \log p\right)x + 2\left(\nabla^2 \log p\right)C\nabla \log p + \nabla\left(\nabla \cdot C\nabla \log p\right)\right)\\
=& \frac{1}{2}\left(s + (\nabla s)x + 2(\nabla s)s + C\nabla(\nabla \cdot s)\right).
\end{aligned}
$$

Thus,

$$
ds(T-t, \overleftarrow{X}_t) = -\frac{1}{2}s(T-t, \overleftarrow{X}_t)dt + \left(\nabla s(T-t, \overleftarrow{X}_t)\right)\sqrt{C}dB_t.
$$

Combining with Itô's formula applied to $e^{-\frac{t}{2}}\left(s(T-t, \overleftarrow{X}_t) + \overleftarrow{X}_t\right)$, we obtain,

$$
\begin{aligned}
& de^{-\frac{t}{2}}\left(s(T-t, \overleftarrow{X}_t) + \overleftarrow{X}_t\right)\\
=& de^{-\frac{t}{2}}\left(s(T-t, \overleftarrow{X}_t)\right) + de^{-\frac{t}{2}}\overleftarrow{X}_t\\
=& e^{-\frac{t}{2}}ds(T-t, \overleftarrow{X}_t) - \frac{1}{2}e^{-\frac{t}{2}}s(T-t, \overleftarrow{X}_t)dt - \frac{1}{2}e^{-\frac{t}{2}}\overleftarrow{X}_t dt + e^{-\frac{t}{2}}d\overleftarrow{X}_t\\
=& e^{-\frac{t}{2}}\nabla s(T-t, \overleftarrow{X}_t)\sqrt{C}d\tilde{B}_t - e^{-\frac{t}{2}}(\frac{1}{2}\overleftarrow{X}_t + s(T-t, \overleftarrow{X}_t))dt + e^{-\frac{t}{2}}d\overleftarrow{X}_t\\
=& e^{-\frac{t}{2}}\left(I + \nabla s(T-t, \overleftarrow{X}_t)\right)\sqrt{C}d\tilde{B}_t.
\end{aligned}
$$

$\square$

**Proof of Theorem 3.4** By Itô's formula and (4), we have, $\forall k \in \{0, 1, ..., N-1\}$ and $\forall t \in [t_k, t_{k+1})$,

$$
de^{\frac{t}{2}}\overleftarrow{X}_t = \frac{1}{2}e^{\frac{t}{2}}\overleftarrow{X}_t dt + e^{\frac{t}{2}}d\overleftarrow{X}_t = e^{\frac{t}{2}}\hat{s}(T-t, \overleftarrow{X}_t)dt + e^{\frac{t}{2}}\sqrt{C}d\tilde{B}_t,
$$

thus,

$$
e^{\frac{t_{k+1}}{2}}\overleftarrow{X}_{t_{k+1}} = e^{\frac{t_k}{2}}\overleftarrow{X}_{t_k} + \int_{t_k}^{t_{k+1}} e^{\frac{t}{2}}\hat{s}(T-t, \overleftarrow{X}_t)dt + \sqrt{e^{t_{k+1}-t_k}}\tilde{z},
$$

where $\tilde{z}_k \sim \mathcal{N}(0, C)$. Now, we couple two processed $\overleftarrow{X}_t$ and $\overleftarrow{Y}_t$ with the same Brownian motion, i.e., let $\bar{z}_k = \tilde{z}_k$ in the sampling scheme (6), then,

$$
\begin{aligned}
e^{\frac{t_{k+1}}{2}}(\overleftarrow{X}_{t_{k+1}} - \overleftarrow{Y}_{t_{k+1}}) =& e^{\frac{t_k}{2}}(\overleftarrow{X}_{t_k} - \overleftarrow{Y}_{t_k}) + \int_{t_k}^{t_{k+1}} \left( e^{\frac{t}{2}}\hat{s}(T-t, \overleftarrow{X}_t) - e^{t-\frac{t_k}{2}}\hat{s}_\theta(T-t_k, \overleftarrow{Y}_{t_k}) \right) dt \\
=& e^{\frac{t_k}{2}}(\overleftarrow{X}_{t_k} - \overleftarrow{Y}_{t_k}) + \int_{t_k}^{t_{k+1}} e^t \left( e^{-\frac{t}{2}}\hat{s}(T-t, \overleftarrow{X}_t) - e^{-\frac{t_k}{2}}\hat{s}(T-t_k, \overleftarrow{X}_{t_k}) \right) dt \\
&+ \int_{t_k}^{t_{k+1}} e^{t-\frac{t_k}{2}} \left( \hat{s}(T-t_k, \overleftarrow{X}_{t_k}) - \hat{s}_\theta(T-t_k, \overleftarrow{X}_{t_k}) \right) dt \\
&+ \int_{t_k}^{t_{k+1}} e^{t-\frac{t_k}{2}} \left( \hat{s}_\theta(T-t_k, \overleftarrow{X}_{t_k}) - \hat{s}_\theta(T-t_k, \overleftarrow{Y}_{t_k}) \right) dt,
\end{aligned}
$$

Recall in Lemma A.3,

$$
e^{-\frac{t}{2}}\hat{s}(T-t, \overleftarrow{X}_t) - e^{-\frac{t_k}{2}}\hat{s}(T-t_k, \overleftarrow{X}_{t_k}) = \int_{t_k}^t e^{-\frac{s}{2}}\nabla\hat{s}(T-s, \overleftarrow{X}_s)\sqrt{C}d\tilde{B}_s,
$$

thus,

$$
\begin{aligned}
\mathbb{E}\left| e^{\frac{t_{k+1}}{2}}(\overleftarrow{X}_{t_{k+1}} - \overleftarrow{Y}_{t_{k+1}}) \right|^2 =& \mathbb{E}\Big| e^{\frac{t_k}{2}}(\overleftarrow{X}_{t_k} - \overleftarrow{Y}_{t_k}) \\
&+ \int_{t_k}^{t_{k+1}} e^{t-\frac{t_k}{2}} \left( \hat{s}_\theta(T-t_k, \overleftarrow{X}_{t_k}) - \hat{s}_\theta(T-t_k, \overleftarrow{Y}_{t_k}) \right) dt \\
&+ \int_{t_k}^{t_{k+1}} e^{t-\frac{t_k}{2}} \left( \hat{s}(T-t_k, \overleftarrow{X}_{t_k}) - \hat{s}_\theta(T-t_k, \overleftarrow{X}_{t_k}) \right) dt \Big|^2 \\
&+ \mathbb{E}\left| \int_{t_k}^{t_{k+1}} e^t \int_{t_k}^t e^{-\frac{s}{2}}\nabla\hat{s}(T-s, \overleftarrow{X}_s)\sqrt{C}d\tilde{B}_s dt \right|^2 \\
:=& I_1^k + I_2^k.
\end{aligned}
$$

By Cauchy's inequality,

$$
\begin{aligned}
I_2^k \leq& \mathbb{E}\left| \int_{t_k}^{t_{k+1}} e^{2t}dt \int_{t_k}^{t_{k+1}} | \int_{t_k}^t e^{-\frac{s}{2}}\nabla\hat{s}(T-s, \overleftarrow{X}_t)\sqrt{C}d\tilde{B}_s|^2 dt \right| \\
=& \frac{e^{2t_{k+1}} - e^{2t_k}}{2} \int_{t_k}^{t_{k+1}} \mathbb{E}| \int_{t_k}^t e^{-\frac{s}{2}}\nabla\hat{s}(T-s, \overleftarrow{X}_s)\sqrt{C}d\tilde{B}_s|^2 dt.
\end{aligned}
$$

By Itô's isometry and (32),

$$
\begin{aligned}
\mathbb{E}| \int_{t_k}^t e^{-\frac{s}{2}}\nabla\hat{s}(T-s, \overleftarrow{X}_s)\sqrt{C}d\tilde{B}_s|^2 \leq& \int_{t_k}^t e^{-s}(L_0 + L_2)^2 e^{-2T+2s}\text{Tr}(C)ds \\
\leq& (L_0 + L_2)^2\text{Tr}(C)e^{-2T}(e^{t_{k+1}} - e^{t_k}).
\end{aligned}
\tag{36}
$$

With $\tau$ satisfying (34), we have,

$$
I_2^k \leq e^3(L_0 + L_2)^2\text{Tr}(C)e^{-2T+3t_k}\tau^2(t_{k+1} - t_k).
\tag{37}
$$

By the mean inequality, for any positive number $f_k$, we have

$$
\begin{aligned}
I_1^k \leq& (1+f_k)\mathbb{E}\left| e^{\frac{t_k}{2}}(\overleftarrow{X}_{t_k} - \overleftarrow{Y}_{t_k}) + \int_{t_k}^{t_{k+1}} e^{t-\frac{t_k}{2}} \left( \hat{s}_\theta(T-t_k, \overleftarrow{X}_{t_k}) - \hat{s}_\theta(T-t_k, \overleftarrow{Y}_{t_k}) \right) dt \right|^2 \\
&+ (1+f_k^{-1})\mathbb{E}\left| \int_{t_k}^{t_{k+1}} e^{t-\frac{t_k}{2}} \left( \hat{s}(T-t_k, \overleftarrow{X}_{t_k}) - \hat{s}_\theta(T-t_k, \overleftarrow{X}_{t_k}) \right) dt \right|^2
\end{aligned}
\tag{38}
$$

By (33) and the condition (34) on $\tau$, the first term of (38) satisfies,

$$\mathbb{E}\left| e^{\frac{t_k}{2}}(\overleftarrow{X}_{t_k} - \overleftarrow{Y}_{t_k}) + \int_{t_k}^{t_{k+1}} e^{t-\frac{t_k}{2}}\left( \hat{s}_\theta(T - t_k, \overleftarrow{X}_{t_k}) - \hat{s}_\theta(T - t_k, \overleftarrow{Y}_{t_k}) \right) dt \right|^2$$

$$\leq \mathbb{E}\left| e^{\frac{t_k}{2}}(\overleftarrow{X}_{t_k} - \overleftarrow{Y}_{t_k}) + \int_{t_k}^{t_{k+1}} e^{t-\frac{t_k}{2}}(L_0 + L_2)e^{-T+t_k}(\overleftarrow{X}_{t_k} - \overleftarrow{Y}_{t_k})dt \right|^2$$

$$\leq \left( 1 + (L_0 + L_2)e^{-T}\int_{t_k}^{t_{k+1}} e^t dt \right)^2 \mathbb{E}|e^{\frac{t_k}{2}}(\overleftarrow{X}_{t_k} - \overleftarrow{Y}_{t_k})|^2$$

$$\leq \left( 1 + \left( 2(L_0 + L_2) + (L_0 + L_2)^2 e\tau \right)e^{-T}(e^{t_{k+1}} - e^{t_k}) \right)\mathbb{E}|e^{\frac{t_k}{2}}(\overleftarrow{X}_{t_k} - \overleftarrow{Y}_{t_k})|^2,$$

Denoting

$$\epsilon_k := \sqrt{\mathbb{E}|s(T - t_k, \overleftarrow{X}_{t_k}) - s_\theta(T - t_k, \overleftarrow{X}_{t_k})|^2},$$

the second term of (38) satisfies,

$$\mathbb{E}\left| \int_{t_k}^{t_{k+1}} e^{t-\frac{t_k}{2}}\left( \hat{s}(T - t_k, \overleftarrow{X}_{t_k}) - \hat{s}_\theta(T - t_k, \overleftarrow{X}_{t_k}) \right) dt \right|^2$$

$$= (e^{t_{k+1}} - e^{t_k})^2 e^{-t_k} \mathbb{E}|s(T - t_k, \overleftarrow{X}_{t_k}) - s_\theta(T - t_k, \overleftarrow{X}_{t_k})|^2$$

$$= (e^{t_{k+1}} - e^{t_k})^2 e^{-t_k} \epsilon_k^2.$$

Now, for all $f_k > 0$, we have,

$$I_1^k \leq (1 + f_k)\left( 1 + \left( 2(L_0 + L_2) + (L_0 + L_2)^2 e\tau \right)e^{-T}(e^{t_{k+1}} - e^{t_k}) \right)\mathbb{E}|e^{\frac{t_k}{2}}(\overleftarrow{X}_{t_k} - \overleftarrow{Y}_{t_k})|^2$$

$$+ (1 + f_k^{-1})(e^{t_{k+1}} - e^{t_k})^2 e^{-t_k} \epsilon_k^2.$$

Let,

$$f_k = e^{-T}(e^{t_{k+1}} - e^{t_k}) \leq e^\tau - 1 \leq e,$$

then, by the condition (34) on $\tau$,

$$(1 + f_k)\left( 1 + \left( 2(L_0 + L_2) + (L_0 + L_2)^2 e\tau \right)e^{-T}(e^{t_{k+1}} - e^{t_k}) \right)\mathbb{E}|e^{\frac{t_k}{2}}(\overleftarrow{X}_{t_k} - \overleftarrow{Y}_{t_k})|^2$$

$$\leq \left( 1 + f_k + (1 + e)\left( 2(L_0 + L_2) + (L_0 + L_2)^2 e\tau \right)e^{-T}(e^{t_{k+1}} - e^{t_k}) \right)\mathbb{E}|e^{\frac{t_k}{2}}(\overleftarrow{X}_{t_k} - \overleftarrow{Y}_{t_k})|^2$$

$$= \left( 1 + \left( 1 + (1 + e)(2(L_0 + L_2) + (L_0 + L_2)^2 e\tau) \right)e^{-T}(e^{t_{k+1}} - e^{t_k}) \right)\mathbb{E}|e^{\frac{t_k}{2}}(\overleftarrow{X}_{t_k} - \overleftarrow{Y}_{t_k})|^2,$$

$$(1 + f_k^{-1})(e^{t_{k+1}} - e^{t_k})^2 e^{-t_k}$$

$$\leq (e + 1)f_k^{-1}(e^{t_{k+1}} - e^{t_k})^2 e^{-t_k}$$

$$= (e + 1)e^T e(t_{k+1} - t_k),$$

thus,

$$I_1^k \leq \left( 1 + \left( 1 + (1 + e)(2(L_0 + L_2) + (L_0 + L_2)^2 e\tau) \right)e^{-T}(e^{t_{k+1}} - e^{t_k}) \right)\mathbb{E}|e^{\frac{t_k}{2}}(\overleftarrow{X}_{t_k} - \overleftarrow{Y}_{t_k})|^2$$

$$+ (e + 1)e^T \epsilon_k^2 e(t_{k+1} - t_k). \tag{39}$$

For simplicity, denote $c = \left(1 + (1+e)(2(L_0 + L_2) + (L_0 + L_2)^2 e\tau)\right)$ and $E(k) = \mathbb{E}|e^{\frac{t_k}{2}}(\overleftarrow{X}_{t_k} - \overleftarrow{Y}_{t_k})|^2$. Combining the estimates (37), (39) for $I_2^k$ and $I_1^k$, we obtain,

$$
\begin{aligned}
E(k+1) \leq & \left(1 + ce^{-T}(e^{t_{k+1}} - e^{t_k})\right)E(k) + (e+1)e^T \epsilon_k^2 e(t_{k+1} - t_k) \\
& + e^3(L_0 + L_2)^2 \mathrm{Tr}(C) e^{-2T+3t_k} \tau^2(t_{k+1} - t_k) \\
\leq & \exp(ce^{-T+t_{k+1}})\Big(E(k)\exp(-ce^{-T+t_k}) \\
& + (e+1)e^T \epsilon_k^2 e(t_{k+1} - t_k) + e^3(L_0 + L_2)^2 \mathrm{Tr}(C) e^{-2T}\tau^2(e^{3t_{k+1}} - e^{3t_k})\Big),
\end{aligned}
$$

where we used $\exp(ce^{-T+t_{k+1}}) \geq 1$. Equivalently, we have

$$
\begin{aligned}
& E(k+1)\exp(-ce^{-T+t_{k+1}}) - E(k)\exp(-ce^{-T+t_k}) \\
\leq & (e+1)e^T \epsilon_k^2 e(t_{k+1} - t_k) + e^3(L_0 + L_2)^2 \mathrm{Tr}(C)e^{-2T}\tau^2(e^{3t_{k+1}} - e^{3t_k}).
\end{aligned}
$$

Summing up over $k$ and using the condition in Assumption 3, we obtain,

$$
\begin{aligned}
& \mathbb{E}|e^{\frac{T}{2}}(\overleftarrow{X}_T - \overleftarrow{Y}_T)|^2 \exp(-c) - \mathbb{E}|(\overleftarrow{X}_0 - \overleftarrow{Y}_0)|^2 \exp(-ce^{-T}) \\
\leq & (e+1)e^{T+1}\sum_{k=0}^{N-1} \epsilon_k^2(t_{k+1} - t_k) + e^3(L_0 + L_2)^2 \mathrm{Tr}(C)e^{-2T}\tau^2(e^{3T} - 1) \\
\leq & (e+1)e^{T+1}\epsilon^2 T + e^3(L_0 + L_2)^2 \mathrm{Tr}(C)e^{-2T}\tau^2(e^{3T} - 1),
\end{aligned}
$$

Denoting $K_1 := \exp\left(4 + (1+e)(2(L_0 + L_2) + (L_0 + L_2)^2 e\tau)\right)$, we have the following bound for $\mathbb{E}|\overleftarrow{X}_T - \overleftarrow{Y}_T|^2$,

$$
\mathbb{E}|\overleftarrow{X}_T - \overleftarrow{Y}_T|^2 \leq K_1 \left(e^{-T}\mathbb{E}|\overleftarrow{X}_0 - \overleftarrow{Y}_0|^2 + \epsilon^2 T + (L_0 + L_2)^2 \mathrm{Tr}(C)\tau^2\right).
$$

Now we pick a $\xi$-optimal coupling of $\overleftarrow{X}_0 \sim \overrightarrow{P}_T$ and $\overleftarrow{Y}_0 \sim \overleftarrow{Q}_0 = \gamma_C$ in the Wasserstein distance, i.e.

$$
\mathbb{E}|\overleftarrow{X}_0 - \overleftarrow{Y}_0|^2 \leq \mathcal{W}_2^2(\overrightarrow{P}_T, \overleftarrow{Q}_0) + \xi,
$$

and obtain,

$$
\mathcal{W}_2^2(\overrightarrow{P}_0, \overleftarrow{Q}_T) \leq \mathbb{E}|\overleftarrow{X}_T - \overleftarrow{Y}_T|^2 \leq K_1(e^{-T}(\mathcal{W}_2^2(\overrightarrow{P}_T, \overleftarrow{Q}_0) + \xi) + \epsilon^2 T + (L_0 + L_2)^2 \mathrm{Tr}(C)\tau^2).
$$

Since $\xi$ is arbitrary, the bound will be,

$$
\mathcal{W}_2^2(\overrightarrow{P}_0, \overleftarrow{Q}_T) \leq K_1(e^{-T}\mathcal{W}_2^2(\overrightarrow{P}_T, \overleftarrow{Q}_0) + \epsilon^2 T + (L_0 + L_2)^2 \mathrm{Tr}(C)\tau^2).
$$

Noticing that,

$$
\mathcal{W}_2^2(\overrightarrow{P}_0, \gamma_C) \leq \mathbb{E}_{\overrightarrow{P}_0 \otimes \gamma_C}|x - y|^2 = \mathbb{E}_{\overrightarrow{P}_0}|x|^2 + \mathbb{E}_{\gamma_C}|y|^2 = M_2 + \mathrm{Tr}(C),
$$

and by Lemma A.2,

$$
\mathcal{W}_2^2(\overrightarrow{P}_T, \overleftarrow{Q}_T) = \mathcal{W}_2^2(\overrightarrow{P}_T, \gamma_C) \leq e^{-T}\mathcal{W}_2^2(\overrightarrow{P}_0, \gamma_C).
$$

We get,

$$
\mathcal{W}_2^2(\overrightarrow{P}_0, \overleftarrow{Q}_T) \leq K_1\left(e^{-2T}(M_2 + \mathrm{Tr}(C)) + \epsilon^2 T + (L_0 + L_2)^2 \mathrm{Tr}(C)\tau^2\right). \tag{40}
$$

*Remark* A.4. For the early stop technique, following the same approach, we can get,

$$
\mathcal{W}_2^2(\overrightarrow{P}_\delta, \overleftarrow{Q}_{T-\delta}) \leq K_1\left(e^{-2T+\delta}(M_2 + \mathrm{Tr}(C)) + \epsilon^2 T + (L_0 + L_2)^2 \mathrm{Tr}(C)\tau^2\right). \tag{41}
$$

### A.4 Result under the Assumption 4'

Here we discuss the relaxation of Assumption 4 and the resulting convergence and complexity bound. The preliminary estimate (31) and (32) in Appendix A.3 still hold. Now we denote,

$$b_k := \|\nabla \hat{s}_\theta(T - t_k, \cdot)\|_\infty \quad \text{and} \quad b = \max_k \{b_k\}$$

Then the estimate (33) should be replaced by,

$$\forall k \in \{0, 1, ..., N-1\}, \quad \|\nabla \hat{s}_\theta(T - t_k, \cdot)\|_\infty \le b_k. \tag{42}$$

Following the same approach in Appendix A.3, we still have, for all $k \in \{0, 1, ..N-1\}$

$$\mathbb{E} \left| e^{\frac{t_{k+1}}{2}} (\overleftarrow{X}_{t_{k+1}} - \overleftarrow{Y}_{t_{k+1}}) \right|^2 = I_1^k + I_2^k.$$

$I_2^k$ still satisfies (37) and $I_1^k$ now satisfies, for all $f_k > 0$,

$$\begin{aligned}
I_1^k \le & \left(1 + (2+b)(1+e)b_k e^{-t_k}\left(e^{t_{k+1}} - e^{t_k}\right) + e^{-T}(e^{t_{k+1}} - e^{t_k})\right) \mathbb{E} |e^{\frac{t_k}{2}}(\overleftarrow{X}_{t_k} - \overleftarrow{Y}_{t_k})|^2 \\
& + (e+1)e^T \epsilon_k^2 e(t_{k+1} - t_k).
\end{aligned}$$

Denote,

$$B_0 = 0 \quad \text{and} \quad B_k = \sum_{i=0}^{k-1} \beta_i(t_{i+1} - t_i), \quad k = 1, 2, ..., N-1.$$

Then,

$$\begin{aligned}
I_1^k \le & \left(1 + e^{-T}\left(e^{t_{k+1}} - e^{t_k}\right) + (2+\beta)(1+e)e(B_{k+1} - B_k)\right) \mathbb{E} |e^{\frac{t_k}{2}}(\overleftarrow{X}_{t_k} - \overleftarrow{Y}_{t_k})|^2 \\
& + (e+1)e^T \epsilon_k^2 e(t_{k+1} - t_k).
\end{aligned}$$

And the recursion for $E(k)$ becomes,

$$\begin{aligned}
& E(k+1) \exp\left(-e^{-T+t_{k+1}} - (2+b)e(1+e)B_{k+1}\right) - E(k) \exp\left(-e^{-T+t_k} - (2+b)e(1+e)B_k\right) \\
& \le (e+1)e^T \epsilon_k^2 e(t_{k+1} - t_k) + e^3(L_0 + L_2)^2 \text{Tr}(C)e^{-2T}\tau^2(e^{3t_{k+1}} - e^{3t_k}).
\end{aligned}$$

Summing up over $k$, the bound for $\mathbb{E}|\overleftarrow{X}_T - \overleftarrow{Y}_T|^2$ is of the same form as the one in Appendix A.3, with the different constant coefficient,

$$\mathbb{E}|\overleftarrow{X}_T - \overleftarrow{Y}_T|^2 \le K_1' \left(e^{-T}\mathbb{E}|\overleftarrow{X}_0 - \overleftarrow{Y}_0|^2 + \epsilon^2 T + (L_0 + L_2)^2 \text{Tr}(C)\tau^2\right),$$

where $K_1' := e^{4+(2+b)e(1+e)B}$. The remaining proof follows the same approach as in Appendix A.3, and ultimately yields,

$$\mathcal{W}_2^2(\overrightarrow{P}_0, \overleftarrow{Q}_T) \le K_1' \left(e^{-2T}(M_2 + \text{Tr}(C)) + \epsilon^2 T + (L_0 + L_2)^2 \text{Tr}(C)\tau^2\right).$$

### A.5 Proof of Theorem 3.8

*Proof.* Recall that,

$$q_\sigma(x) = \int_{\mathbb{R}^d} \exp(-\frac{|x-y|^2}{2\sigma^2}) Q_0(dy) = \int_{B(0,R)} \exp(-\frac{|x-y|^2}{2\sigma^2}) Q_0(dy),$$

$$g(x) = \log q_\sigma(x) + \frac{|x|^2}{2\sigma^2}.$$

Fixing $x$, direct computation shows,

$$
\begin{aligned}
\nabla g(x) =& \nabla \log q_\sigma(x) + \frac{x}{\sigma^2} = \frac{\int_{B(0,R)} (\frac{y-x}{\sigma^2}) \exp(-\frac{|x-y|^2}{2\sigma^2}) Q_0(dy)}{\int_{B(0,R)} \exp(-\frac{|x-y|^2}{2\sigma^2}) Q_0(dy)} + \frac{x}{\sigma^2} \\
=& \frac{\int_{B(0,R)} (\frac{y}{\sigma^2}) \exp(-\frac{|x-y|^2}{2\sigma^2}) Q_0(dy)}{\int_{B(0,R)} \exp(-\frac{|x-y|^2}{2\sigma^2}) Q_0(dy)}.
\end{aligned}
$$

Taking absolute value, we get,

$$
|\nabla g(x)| = |\nabla \log q_\sigma(x) + \frac{x}{\sigma^2}| \le \frac{R}{\sigma^2}.
$$

For any unit direction $z$,

$$
\begin{aligned}
&\nabla_z \cdot \nabla_z (\log q_\sigma + \frac{|x|^2}{2\sigma^2}) \\
=& \frac{1}{\sigma^2} \nabla_z \left( \frac{\int_{B(0,R)} (y \cdot z) \exp(-\frac{|x-y|^2}{2\sigma^2}) Q_0(dy)}{\int_{B(0,R)} \exp(-\frac{|x-y|^2}{2\sigma^2}) Q_0(dy)} \right) \\
=& \frac{1}{\sigma^2} \left( \frac{\int_{B(0,R)} (y \cdot z)(-\frac{x-y}{\sigma^2} \cdot z) \exp(-\frac{|x-y|^2}{2\sigma^2}) Q_0(dy)}{\int_{B(0,R)} \exp(-\frac{|x-y|^2}{2\sigma^2}) Q_0(dy)} \right. \\
& \left. - \frac{\int_{B(0,R)} (y \cdot z) \exp(-\frac{|x-y|^2}{2\sigma^2}) Q_0(dy) \int_{B(0,R)} (-\frac{x-y}{\sigma^2} \cdot z) \exp(-\frac{|x-y|^2}{2\sigma^2}) Q_0(dy)}{(\int_{B(0,R)} \exp(-\frac{|x-y|^2}{2\sigma^2}) Q_0(dy))^2} \right) \\
=& \frac{1}{\sigma^4} \left( \frac{\int_{B(0,R)} (y \cdot z)^2 \exp(-\frac{|x-y|^2}{2\sigma^2}) Q_0(dy)}{\int_{B(0,R)} \exp(-\frac{|x-y|^2}{2\sigma^2}) Q_0(dy)} - \frac{(\int_{B(0,R)} (y \cdot z) \exp(-\frac{|x-y|^2}{2\sigma^2}) Q_0(dy))^2}{(\int_{B(0,R)} \exp(-\frac{|x-y|^2}{2\sigma^2}) Q_0(dy))^2} \right).
\end{aligned}
$$

Taking absolute values of the inner terms, we have,

$$
\left| \frac{\int_{B(0,R)} (y \cdot z)^2 \exp(-\frac{|x-y|^2}{2\sigma^2}) Q_0(dy)}{\int_{B(0,R)} \exp(-\frac{|x-y|^2}{2\sigma^2}) Q_0(dy)} \right| \le \frac{\int_{B(0,R)} R^2 \exp(-\frac{|x-y|^2}{2\sigma^2}) Q_0(dy)}{\int_{B(0,R)} \exp(-\frac{|x-y|^2}{2\sigma^2}) Q_0(dy)} \le R^2,
$$

and

$$
\left| \frac{\int_{B(0,R)} (y \cdot z) \exp(-\frac{|x-y|^2}{2\sigma^2}) Q_0(dy)}{\int_{B(0,R)} \exp(-\frac{|x-y|^2}{2\sigma^2}) Q_0(dy)} \right|^2 \le \left| \frac{\int_{B(0,R)} R \exp(-\frac{|x-y|^2}{2\sigma^2}) Q_0(dy)}{\int_{B(0,R)} \exp(-\frac{|x-y|^2}{2\sigma^2}) Q_0(dy)} \right| \le R^2.
$$

Finally, by taking the absolute value, we obtain,

$$
|\nabla_z \cdot \nabla_z g(x)| = |\nabla_z \cdot \nabla_z (\log q_\sigma + \frac{|x|^2}{2\sigma^2})| \le \frac{2R^2}{\sigma^4}.
$$

Thus,

$$
\|\nabla^2 g\|_\infty \le \frac{2R^2}{\sigma^4}.
$$

$\square$

## A.6  Proof of Corollary 3.9

*Proof.* Recall that, in Corollary 3.9, we have,

$$
\begin{cases}
C = I_d, \\
A = (1 - e^{-\delta}) I_d, \\
\bar{A}_t = (1 - e^{-t}) I_d.
\end{cases}
$$

And the distribution of the forward OU process at time $\delta$ is given by

$$\overrightarrow{P}_\delta = \mathcal{N}\big(0, (1 - e^{-\delta})I_d\big) * P_{0,\delta},$$

where $P_{0,\delta} := Law(e^{-\frac{\delta}{2}}X_0)$, $X_0 \sim \overrightarrow{P}_0$ and $P_{0,\delta}$ satisfies $\mathrm{Diam}(\mathrm{Supp}\, P_{0,\delta}) \le e^{-\frac{\delta}{2}}R$. Then, by Theorem 3.8, the corresponding $h(x) = \log p_\delta(x) + \frac{|x|^2}{2(1 - e^\delta)}$ satisfies,

$$|\nabla h|_\infty \le \frac{Re^{-\frac{\delta}{2}}}{1 - e^{-\delta}}, \quad \|\nabla^2 h\| \le 2\frac{R^2 e^{-\delta}}{(1 - e^{-\delta})^2}.$$

The constants then can be computed directly,

$$\begin{cases} K = \sup_{t \ge \delta} \|A\bar{A}_t^{-1}\| = 1, \\ L_0 = K^2(\|\nabla h\|_\infty + |\nabla h|_\infty^2) \le 3\frac{R^2 e^{-\delta}}{(1 - e^{-\delta})^2}, \\ L_1 = K|\nabla h|_\infty \le \frac{Re^{-\frac{\delta}{2}}}{1 - e^{-\delta}}. \end{cases}$$

Then, by Corollary 3.2, for all $t \ge \delta$,

$$\begin{aligned} \|\nabla^2 \log p_t(x) + \bar{A}_t^{-1}\|_\infty &\le L_0 e^{-(t-\delta)} = 3\frac{R^2}{(1 - e^{-\delta})^2}e^{-t}, \\ |\nabla^2 \log p_t(x) + \bar{A}_t^{-1}x|_\infty &\le L_1 e^{-\frac{t-\delta}{2}} = \frac{R}{1 - e^{-\delta}}e^{-\frac{t}{2}}. \end{aligned} \tag{43}$$

$\square$

## A.7 Proof of Theorem 3.10

*Proof.* The proof follows the same procedure as in Appendix A.3; it suffices to substitute the corresponding values of the constants. By Corollary 3.9,

$$\begin{cases} C = I_d, \\ L_0 = 3\frac{R^2}{(1 - e^{-\delta})^2}, \\ L_2 = \frac{1}{1 - e^{-\delta}}. \end{cases}$$

Noticing that under bounded-support assumption,

$$M_2 = \mathbb{E}|\overrightarrow{X}_0|^2 \le R^2,$$

and the step size $\tau$ is sufficiently small to satisfy

$$(L_0 + L_2)e\tau \le 1.$$

By Remark A.4, we get,

$$\begin{aligned} \mathcal{W}_2(\overrightarrow{P}_\delta, \overleftarrow{Q}_{T-\delta}) &\le \exp\big(4 + 3(1 + e)(L_0 + L_2)\big)\big(e^{-2T+\delta}\big(R^2 + d\big) + \epsilon^2 T + (L_0 + L_2)^2 d\tau^2\big) \\ &\le \exp\big(4 + 3(1 + e)K_2\big)\big(e^{-2T+\delta}\big(R^2 + d\big) + \epsilon^2 T + K_2^2 d\tau^2\big), \end{aligned}$$

where $K_2 = 3\frac{R^2}{(1 - e^{-\delta})^2} + \frac{1}{1 - e^{-\delta}}$. $\square$

## A.8 Proof of Theorem 3.13

*Proof.* We first validate that the posterior follows Gaussian tail Assumption 2 with $A = C$, $h(x) = -\frac{|G(x) - y|_\Sigma^2}{2}$, and,

$$\begin{aligned} |\sqrt{C}\nabla h(x)| &= |\sqrt{C}\nabla G(x)\Sigma^{-1}(G(x) - y)| \le \|C\|^{\frac{1}{2}}(|G|_\infty + |y|)\|\Sigma\|^{-1}\|\nabla G\|_\infty, \\ \|C\nabla^2 h(x)\| &= \|C\nabla^2 G(x)\Sigma^{-1}(G(x) - y) + C\nabla G(x)\Sigma^{-1}\nabla G(x)^T\| \\ &\le \|C\|\|\Sigma\|^{-1}\Big(\|\nabla^2 G\|_\infty(|G|_\infty + |y|) + \|\nabla G\|_\infty^2\Big). \end{aligned}$$

Then, it suffices to substitute the corresponding values of the constants in Appendix A.3,

$$
\begin{aligned}
K =& \max\{1, \|AC^{-1}\|\} = 1, \\
L_0 =& K^2\big(\|C\nabla^2 h\|_\infty + |\sqrt{C}\nabla h|_\infty^2\big) \\
\leq& \|C\|\|\Sigma\|^{-1}\bigg(\Big(\|\nabla^2 G\|_\infty(|G|_\infty + |y|) + \|\nabla G\|_\infty^2\Big) + \|\Sigma\|^{-1}\|\nabla G\|_\infty^2\Big(|G|_\infty + |y|\Big)^2\bigg), \\
L_2 =& \max\{\|I - CA^{-1}\|, \|AC^{-1} - I\|\} = 0.
\end{aligned}
$$

Now following the proof in Appendix A.3, we have,

$$
\mathcal{W}_2^2\big(\overrightarrow{P}_0(\cdot; y), \overleftarrow{Q}_T(\cdot; y)\big) \leq K_3\Big(e^{-2T}\big(M_2 + \mathrm{Tr}(C)\big) + \epsilon^2 T + k_3^2 \mathrm{Tr}(C)\tau^2\Big),
$$

where

$$
K_3 = e^{4+3(1+e)k_3},
$$

$$
k_3 = \|C\|\|\Sigma\|^{-1}\bigg(\Big(\|\nabla^2 G\|_\infty(|G|_\infty + |y|) + \|\nabla G\|_\infty^2\Big) + \|\Sigma\|^{-1}\|\nabla G\|_\infty^2\Big(|G|_\infty + |y|\Big)^2\bigg).
$$

$\square$

# B  Theories towards the generative diffusion model in infinite dimension

## B.1  Motivating example towards the infinite dimensional result

We consider the following target distribution,

$$
p_0(x) = \frac{1}{Z}\prod_{i=1}^{d}\Big(\frac{1}{2}\exp(-\frac{|x - \sqrt{C_i}|^2}{2C_i}) + \frac{1}{2}\exp(-\frac{|x + \sqrt{C_i}|^2}{2C_i})\Big),
$$

where $C$ is an $d \times d$ matrix diagonal matrix with $\{C_i\}_i$ as diagonal entries. Denote the case $d = 1$ and $C_1 = 1$ as,

$$
p_0^1(x) = \frac{1}{2\sqrt{2\pi}}\Big(\exp(-\frac{(x-1)^2}{2}) + \exp(-\frac{(x+1)^2}{2})\Big),
$$

and if one considers to apply the forward process (1), one can get the distribution at time $t$,

$$
p_t^1(x) = \frac{1}{2\sqrt{2\pi}}\Big(\exp(-\frac{(x - e^{-\frac{t}{2}})^2}{2}) + \exp(-\frac{(x + e^{-\frac{t}{2}})^2}{2})\Big).
$$

Simple calculation shows,

$$
\mathcal{W}_2\big(p_0, \mathcal{N}(0, C)\big) \leq \sqrt{2\mathrm{Tr}(C)}, \quad KL\big(p_0 || \mathcal{N}(0, C)\big) = d \cdot KL\big(p_0^1 || \mathcal{N}(0, 1)\big). \tag{44}
$$

And if one considers applying the forward process (1), one can also show that,

$$
\mathcal{W}_2\big(p_t, N(0, C)\big) \leq e^{-\frac{t}{2}}\sqrt{2\mathrm{Tr}(C)}, \quad KL\big(p_t || \mathcal{N}(0, C)\big) = d \cdot KL\big(p_t^1 || \mathcal{N}(0, 1)\big). \tag{45}
$$

From (44) and (45), we observe that *when increasing the dimension of $C$ while keeping* $\mathrm{Tr}(C)$ *fixed,* the Wasserstein-2 bounds only scale with the trace of $C$ while the KL bounds scale with $d$.

## B.2 Defining diffusion model in infinite dimension

Now we consider a separable Hilbert space $H$ with inner product $\langle \cdot, \cdot \rangle_H$, we denote by $\mathcal{L}(H)$ the space of bounded linear operators on $H$. The forward process in $H$ has the same form as (1),

$$d\overrightarrow{X}_t = -\frac{1}{2}\overrightarrow{X}_t dt + \sqrt{C}dB_t, \quad 0 \le t \le T, \tag{46}$$

where $C$ becomes a degenerate positive trace-class. We still denote the marginal distributions of $\overrightarrow{X}_t$ by $\overrightarrow{P}_t$, which converges to the stationary distribution $\mathcal{N}(0, C)$ as $t \to \infty$ (Da Prato & Zabczyk (2014), Theorem 11.11). After time reversal $\overleftarrow{X}_t = \overrightarrow{X}_{T-t}$, $\overleftarrow{X}_t$ satisfies the backward SDE (Pidstrigach et al. (2024), Theorem 1),

$$d\overleftarrow{X}_t = \left(\frac{1}{2}\overleftarrow{X}_t + s(T - t, \overleftarrow{X}_t)\right)dt + \sqrt{C}d\tilde{B}_t, \tag{47}$$

where score function $s(t, x)$ is defined as:

$$s(t, x) := -\frac{1}{1 - e^{-t}}\mathbb{E}[\overrightarrow{X}_t - e^{-\frac{t}{2}}\overrightarrow{X}_0 | \overrightarrow{X}_t = x], \tag{48}$$

which is almost surely continuous in $t$ with respect to the norm $\|\|_H$ and equal to $C\nabla \log p_t(x)$ when $H = \mathbb{R}^d$. We consider $s_\theta(t, x)$ to approximate the score operator in $H$ (48) and the scheme (6) for the sampling process. We now introduce the infinite-dimensional Gaussian tail assumption.

**Assumption 6.** *The initial data distribution $\overrightarrow{P}_0$ has finite second moment $M_2$ and has a Gaussian tail, i.e.,*

$$d\overrightarrow{P}_0(x) \propto \exp\left(h(x)\right)d\mathcal{N}(0, A)(x), \tag{49}$$

*where $A$ is a degenerate positive trace-class operator that is simultaneously diagonalizable with $C$, $A$ and $C$ correspond to the same Cameron–Martin space, and both $AC^{-1}$ and $A^{-1}C$ are bounded linear maps. The function $h$ is two times differentiable and satisfies,*

$$\begin{aligned}
\|\sqrt{C}\nabla h\|_{H,\infty} &:= \sup_x \|\nabla\sqrt{C}h(x)\|_H < \infty, \\
\|C\nabla^2 h\|_{\mathcal{L}(H),\infty} &:= \sup_x \|C\nabla^2 h(x)\|_{\mathcal{L}(H)} < \infty.
\end{aligned} \tag{50}$$

To extend our Corollary 3.2 and Theorem 3.4 to the infinite-dimensional case, we first follow the approach in Appendix E of (Pidstrigach et al., 2024) that projects $H$ onto a finite-dimensional subspace $H^D$, and approximates the infinite-dimensional case via the results established on $H^D$.

Suppose $C$ ($A$, correspondingly) has an orthonormal basis $e_i$ of eigenvectors and corresponding non-negative eigenvalues $c_i \ge 0$ ($a_i \ge 0$, correspondingly), we define the linear span of the first $D$ eigenvectors as

$$H^D = \mathrm{span}\{e_1, e_2, \dots, e_D\}.$$

Let $P^D : H \to H^D$ be the orthogonal projection onto $H^D$. We define the finite-dimensional approximations of $\mu_{data}$ by $\mu_{data}^D = P^D{}_\#(\mu_{data})$ and discretize the forward process by $\overrightarrow{X_t^D} = P^D\overrightarrow{X}_t$, then $(\overrightarrow{X_t^D})_{t\ge 0}$ will satisfy,

$$d\overrightarrow{X_t^D} = -\frac{1}{2}\overrightarrow{X_t^D}dt + \sqrt{P^D C P^D}dB_t. \tag{51}$$

We denote the marginal distribution of $\overrightarrow{X_t^D}$ by $\overrightarrow{P_t^D}$, and its density by $p_t^D$. The corresponding backward process will be,

$$d\overleftarrow{X_t^D} = \left(\frac{1}{2}\overleftarrow{X_t^D} + s^D(T - t, \overleftarrow{X_t^D})\right)dt + \sqrt{P^D C P^D}d\tilde{B}_t, \tag{52}$$

where $s^D(t, x) = P^D\mathbb{E}[s(t, \overrightarrow{X}_t)|P^D\overrightarrow{X}_t = x] = C^D\nabla \log p_t^D(x)$.

Then we show the projected distribution still follows the Gaussian tail assumption in finite dimensions, summarized as follows.

**Lemma B.1.** *The projected initial distribution $\overrightarrow{P_t^D}$ satisfies the finite-dimensional Gaussian tail Assumption 2,*

$$d\overrightarrow{P_0^D}(x^D) \propto \exp\left(h^D(x^D)\right)d\mathcal{N}(0, A^D)(x^D),$$

*where*

$$
\begin{aligned}
|\sqrt{C^D}\nabla h^D|_\infty &\le \|\sqrt{C}\nabla h\|_{H,\infty}, \\
\|C^D\nabla^2 h^D\|_\infty &\le \|C\nabla^2 h\|_{\mathcal{L}(H),\infty} + \|\sqrt{C}\nabla h\|_{H,\infty}^2.
\end{aligned}
\tag{53}
$$

*Proof.* We define $P^{D+1:\infty} : H \to H^{D+1:\infty}$ be the projection onto $\text{span}\{e_{D+1}, e_{D+2}, \cdots\}$, covariance operator $A^{D+1:\infty} := P^{D+1:\infty} A P^{D+1:\infty}$ and $x^{D+1:\infty} := P^{D+1:\infty} x$. The term $\exp(h^D(x^D))$ is given by,

$$
\begin{aligned}
\exp(h^D(x^D)) &= \int \exp\left(h(x^D, x^{D+1:\infty})\right)d\mathcal{N}(0, A^{D+1:\infty})(x^{D+1:\infty}) \\
&= \mathbb{E}_{\mathcal{N}(0,A)}[\exp\left(h(X)\right)|X^D = x^D].
\end{aligned}
$$

Then, the gradient is given by,

$$
\begin{aligned}
\nabla h^D(x^D) &= \frac{\int \nabla_{x^D} h(x^D, x^{D+1:\infty}) \exp\left(h(x^D, x^{D+1:\infty})\right)d\mathcal{N}(0, A^{D+1:\infty})(x^{D+1:\infty})}{\int \exp\left(h(x^D, x^{D+1:\infty})\right)d\mathcal{N}(0, A^{D+1:\infty})(x^{D+1:\infty})} \\
&= \mathbb{E}_{\tilde{P}_0}[\nabla_{x^D} h(X)|X^D = x^D],
\end{aligned}
$$

where $d\tilde{P}_0(x) \propto \exp\left(h(x)\right)d\mathcal{N}(0, A)(x)$, and the Hessian is given by,

$$
\begin{aligned}
\nabla^2 h^D(x^D) =& \frac{\int \nabla_{x^D}^2 h(x^D, x^{D+1:\infty}) \exp\left(h(x^D, x^{D+1:\infty})\right)d\mathcal{N}(0, A^{D+1:\infty})(x^{D+1:\infty})}{\int \exp\left(h(x^D, x^{D+1:\infty})\right)d\mathcal{N}(0, A^{D+1:\infty})(x^{D+1:\infty})} \\
&- \left(\frac{\int \nabla_{x^D} h(x^D, x^{D+1:\infty}) \exp\left(h(x^D, x^{D+1:\infty})\right)d\mathcal{N}(0, A^{D+1:\infty})(x^{D+1:\infty})}{\int \exp\left(h(x^D, x^{D+1:\infty})\right)d\mathcal{N}(0, A^{D+1:\infty})(x^{D+1:\infty})}\right)^{\otimes 2} \\
=& \mathbb{E}_{\tilde{P}_0}[\nabla_{x^D}^2 h(X)|X^D = x^D] - (\mathbb{E}_{\tilde{P}_0}[\nabla_{x^D} h(X)|X^D = x^D])^{\otimes 2}.
\end{aligned}
$$

Taking the absolute values, we get,

$$
\begin{aligned}
|\sqrt{C^D}\nabla h^D(x^D)| &\le |\sqrt{C^D}\nabla_{x^D} h|_\infty \le \|\sqrt{C}\nabla h\|_{H,\infty}, \\
\|C^D\nabla^2 h^D(x^D)\| &\le \|C^D\nabla_{x^D}^2 h\|_\infty + |\sqrt{C^D}\nabla_{x^D} h|_\infty^2 \le \|C\nabla^2 h\|_{\mathcal{L}(H),\infty} + \|\sqrt{C}\nabla h\|_{H,\infty}^2.
\end{aligned}
$$

Thus,

$$
\begin{aligned}
|\sqrt{C^D}\nabla h^D|_\infty &\le \|\sqrt{C}\nabla h\|_{H,\infty}, \\
\|C^D\nabla^2 h^D\|_\infty &\le \|C\nabla^2 h\|_{\mathcal{L}(H),\infty} + \|\sqrt{C}\nabla h\|_{H,\infty}^2.
\end{aligned}
$$

$\square$

Define,

$$
\begin{aligned}
\bar{A}_t &:= Ae^{-t} + C(1 - e^{-t}), \\
\bar{A}_t^D &:= A^D e^{-t} + C^D(1 - e^{-t}),
\end{aligned}
$$

and

$$
\begin{aligned}
\tilde{s}^D(t, x) &:= s^D(t, x) + C^D(\bar{A}_t^D)^{-1}x, \\
\tilde{s}(t, x) &:= s(t, x) + C\bar{A}_t^{-1}x.
\end{aligned}
\tag{54}
$$

Since the projected initial distribution $\overrightarrow{P_0^D}$ satisfies the finite-dimensional Gaussian tail assumption (Assumption 2), Corollary 3.2 implies that there exist constants $L_0^D, L_1^D \ge 0$ such that,

$$\|\nabla\tilde{s}^D(t, \cdot)\|_{\mathcal{L}(H^D),\infty} \le L_0^D e^{-t}, \quad \|\tilde{s}^D(t, \cdot)\|_{H^D,\infty} \le L_1^D e^{-\frac{t}{2}}, \quad \forall t \ge 0.$$

Here the constant $L_0^D$ and $L_1^D$ is given by,

$$\begin{cases} K^D = \max\{1, \|A^D(C^D)^{-1}\|\} \le \max\{1, \|AC^{-1}\|\} := K, \\ L_0^D = (K^D)^2 \big( \|C^D \nabla^2 h^D\|_\infty + |\sqrt{C^D}\nabla h^D|_\infty^2 \big) \le K^2 \big( \|C\nabla^2 h\|_{\mathcal{L}(H),\infty} + 2\|\sqrt{C}\nabla h\|_{H,\infty}^2 \big), \\ L_1^D = K^D \|C^D\|^{1/2} |\sqrt{C^D}\nabla h^D|_\infty \le K\|C\|^{\frac{1}{2}} \|\sqrt{C}\nabla h\|_{H,\infty}. \end{cases}$$

By (Pidstrigach et al., 2024) lemma 16 (listed as Proposition B.4), we have

$$\tilde{s}^D(t, \overrightarrow{X_t^D}) \to \tilde{s}(t, \overrightarrow{X}_t) = s(t, \overrightarrow{X}_t) + C\bar{A}_t^{-1}\overrightarrow{X}_t, \quad \text{both } a.s. \text{ and } L_2, \quad \text{as } D \to \infty. \tag{55}$$

Thus, we obtain the following regularity estimation for the modified score,

**Theorem B.2.** *Under the infinite dimensional Gaussian tail Assumption 6, the modified score $\tilde{s}$ defined by (48) and (54) satisfies,*

$$\|\nabla \tilde{s}(t, \cdot)\|_{\mathcal{L}(H),\infty} \le \tilde{L}_0 e^{-t}, \quad \|\tilde{s}(t, x)\|_{H,\infty} \le \tilde{L}_1 e^{-\frac{t}{2}}, \quad \forall t > 0. \tag{56}$$

*where*

$$\begin{cases} K = \max\{1, \|AC^{-1}\|\}, \\ \tilde{L}_0 = K^2 \big( \|C\nabla^2 h\|_{\mathcal{L}(H),\infty} + 2\|\sqrt{C}\nabla h\|_{H,\infty}^2 \big), \\ \tilde{L}_1 = K\|C\|^{\frac{1}{2}} \|\sqrt{C}\nabla h\|_{H,\infty}. \end{cases}$$

(56) pertains to Corollary 3.2 formulated in the infinite-dimensional case. To derive an error bound, we also need Assumption 3 and 4 under an infinite-dimensional setting, listed as follows.

**Assumption 7.** *For each time discretization point $t_k$, $0 \le k \le N-1$, the approximated score $\tilde{s}_\theta$ satisfies,*

$$\|\nabla \tilde{s}_\theta(T - t_k, \cdot)\|_{\mathcal{L}(H),\infty} \le \tilde{L}_0 e^{-T + t_k},$$

$$\frac{1}{T} \sum_{k=0}^{N-1} (t_{k+1} - t_k) \mathbb{E}\|s(T - t_k, \overrightarrow{X}_{T-t_k}) - s_\theta(T - t_k, \overrightarrow{X}_{T-t_k})\|_H^2 \le \epsilon^2.$$

To measure the distance between the samples and the target distribution, we introduce the Wasserstein-2-distance on the Hilbert space $H$,

$$\mathcal{W}_2(\mu, \nu) = \left( \inf_\pi \int \|x - y\|_H^2 d\pi(x, y) \right)^{1/2},$$

where $\pi$ runs over all measures on $H \times H$ with marginals $\mu$ and $\nu$.

We now present the convergence bound aligned with Theorem 3.4 in the infinite-dimensional setting. In particular, the main additional ingredients consist of extension of Lemma A.3 and the approximation of $I_k^2$ to infinite dimensions, while the remaining arguments follow the same lines as in the finite-dimensional case.

**Theorem B.3.** *Suppose Assumptions 6 and 7 hold under the infinite infinite-dimensional setting. Then, sampling via scheme (6) with step size $\tau \le 1$, we obtain the following convergence bound,*

$$\mathcal{W}_2^2(\overrightarrow{P}_0, \overleftarrow{Q}_T) \le \tilde{K}_1 \Big( e^{-2T} (M_2 + \text{Tr}(C)) + \epsilon^2 T + (\tilde{L}_0 + L_2)^2 \text{Tr}(C)\tau^2 \Big), \tag{57}$$

*where $\tilde{K}_1 = e^{4 + (e+1)\big(2(\tilde{L}_0 + L_2) + (\tilde{L}_0 + L_2)^2 e\tau\big)}$.*

*Proof.* First, we list the preliminary estimates as in Appendix A.3. Denote

$$\hat{s}(t, x) := s(t, x) + x, \quad \hat{s}_\theta(t, x) := s_\theta(t, x) + x$$

and

$$L_2 := \max\{\|I - CA^{-1}\|, \|AC^{-1} - I\|\} = \sup_t e^t \|I - C\bar{A}_t^{-1}\|.$$

By Theorem B.2 and Assumption 7, we have,

$$\forall t \geq 0, \quad \|\nabla \hat{s}(t, \cdot)\|_{\mathcal{L}(H), \infty} \leq (\tilde{L}_0 + L_2) e^{-t},$$
$$\forall k \in \{0, 1, ..., N-1\}, \quad \|\nabla \hat{s}_\theta(T - t_k, \cdot)\|_{\mathcal{L}(H), \infty} \leq (\tilde{L}_0 + L_2) e^{-T + t_k}.$$

The proof proceeds in the same manner as in Appendix A.3. However, Lemma A.3 is not available here, since in the infinite-dimensional setting there is no probability density in the sense of Lebesgue measure. Therefore, we make appropriate adjustments at the points where Lemma A.3 would otherwise be used. By Theorem B.2, $\hat{s}(T - t, \overleftarrow{X}_t)$ satisfies,

$$
\begin{aligned}
\mathbb{E}\|\hat{s}(T - t, \overleftarrow{X}_t)\|_H^2 &= \mathbb{E}\|\tilde{s}(T - t, \overleftarrow{X}_t) + (I - C\bar{A}_{T-t}^{-1})\overleftarrow{X}_t\|^2 \\
&\leq 2\mathbb{E}\|\tilde{s}(T - t, \overleftarrow{X}_t)\|_H^2 + 2\mathbb{E}\|(I - C\bar{A}_{T-t}^{-1})\overleftarrow{X}_t\|^2 \\
&\leq 2\tilde{L}_1^2 e^{-(T-t)} + 2L_2^2 e^{-2(T-t)} \mathbb{E}\|\overleftarrow{X}_t\|^2 \\
&\leq 2\tilde{L}_1^2 e^{-(T-t)} + 2L_2^2 e^{-2(T-t)} \max\{M_2, \operatorname{Tr}(C)\} \\
&< \infty.
\end{aligned}
$$

Thus, by Proposition B.4, $\hat{s}^D(T - t, \overleftarrow{X}_t^D) := s^D(T - t, \overleftarrow{X}_t^D)$ converge to $\hat{s}(T - t, \overleftarrow{X}_t)$ in $L^2$. By Lemma A.3, $\forall D \in \mathcal{N}^+, e^{-\frac{t}{2}} \hat{s}^D(T - t, X_t^D)$ is a continuous time martingale. Thus, by Doob's $L^2-$inequality, for any $D, N \in \mathcal{N}^+$

$$\mathbb{E}[\sup_{0 \leq t \leq T} |e^{-\frac{t}{2}} \hat{s}^D(T - t, \overleftarrow{X}_t^D) - e^{-\frac{t}{2}} \hat{s}^N(T - t, \overleftarrow{X}_t^N)|^2] \leq 4e^{-T} \mathbb{E}|\hat{s}^D(0, \overleftarrow{X}_T^D) - \hat{s}^N(0, \overleftarrow{X}_T^N)|^2.$$

Since the right side is Cauchy, the left side is also Cauchy, i.e., continuous time martingales $\{e^{-\frac{t}{2}} s^D(T - t, \overleftarrow{X}_t^D)\}_D$ form a Cauchy sequence under the norm,

$$\|s\| = \mathbb{E}[\sup_{t \in [0, T]} |s_t|^2].$$

The continuous martingales are closed with respect to this norm (Karatzas & Shreve (2014), Section 1.3), so $e^{-\frac{t}{2}} s(T - t, \overleftarrow{X}_t)$ is also a continuous time martingale.

Now we follow the approach in Appendix A.3. Coupling $\overleftarrow{X}_t$ and $\overleftarrow{Y}_t$ with the same Brownian motion, and using the fact that $e^{-\frac{t}{2}} s(T - t, \overleftarrow{X}_t)$ is a martingale, we obtain,

$$\mathbb{E} \left| e^{\frac{t_{k+1}}{2}} (\overleftarrow{X}_{t_{k+1}} - \overleftarrow{Y}_{t_{k+1}}) \right|^2 = \tilde{I}_1^k + \tilde{I}_2^k,$$

where

$$
\begin{aligned}
\tilde{I}_1^k = \mathbb{E} \Bigg| e^{\frac{t_k}{2}} (\overleftarrow{X}_{t_k} - \overleftarrow{Y}_{t_k}) &+ \int_{t_k}^{t_{k+1}} e^{t - \frac{t_k}{2}} \left( \hat{s}_\theta(T - t_k, \overleftarrow{X}_{t_k}) - \hat{s}_\theta(T - t_k, \overleftarrow{Y}_{t_k}) \right) dt \\
&+ \int_{t_k}^{t_{k+1}} e^{t - \frac{t_k}{2}} \left( \hat{s}(T - t_k, \overleftarrow{X}_{t_k}) - \hat{s}_\theta(T - t_k, \overleftarrow{X}_{t_k}) \right) dt \Bigg|^2,
\end{aligned}
$$

and

$$\tilde{I}_2^k = \mathbb{E} \left| \int_{t_k}^{t_{k+1}} e^t \left( e^{-\frac{t}{2}} \hat{s}(T - t, \overleftarrow{X}_t) - e^{-\frac{t_k}{2}} \hat{s}(T - t_k, \overleftarrow{X}_{t_k}) \right) dt \right|^2.$$

For $\{e^{-\frac{t}{2}}s^D(T-t,\overset{\leftarrow}{X_t^D})\}_D$, using the estimate in finite-dimensional case for $I_2^k$ in Appendix A.3, we have,

$$\mathbb{E}\left|\int_{t_k}^{t_{k+1}} e^t \left(e^{-\frac{t}{2}}\hat{s}^D(T-t,\overset{\leftarrow}{X_t^D}) - e^{-\frac{t_k}{2}}\hat{s}(T-t_k,\overset{\leftarrow}{X_{t_k}^D})\right) dt\right|^2$$
$$\leq e^3(\tilde{L}_0+L_2)^2\mathrm{Tr}(C)e^{-2T+3t_k}\tau(t_{k+1}-t_k).$$

Since as $D\to\infty$, the $L^2$ convergence of $e^{-\frac{t}{2}}\hat{s}^D(T-t,\overset{\leftarrow}{X_t^D})$ to $e^{-\frac{t}{2}}\hat{s}(T-t,\overset{\leftarrow}{X_t})$ is uniform in time, we obtain,

$$\tilde{I}_k^2 \leq e^3(\tilde{L}_0+L_2)^2\mathrm{Tr}(C)e^{-2T+3t_k}\tau(t_{k+1}-t_k).$$

At this stage, we have modified the parts of the proof in Appendix A.3 that rely on Lemma A.3 and obtained analogous results. The remaining steps follow directly from Appendix A.3, where the corresponding formulas naturally extend to the infinite-dimensional setting. $\qquad\square$

Here, we list the lemma used to extend our analysis to infinite dimensions.

**Proposition B.4.** *(Pidstrigach et al. (2024), Lemma 16) Let $H$ be a separable Hilbert space and $Z,\tilde{Z}$ be two random variables taking values in $H$. Let $e_i$ be an orthonormal basis of $H$. Denote by $H^D = \mathrm{span}\{e_1,e_2,\dots,e_D\}$ and by $P^D$ the projection onto $H^D$. Furthermore, let $Z^D = P^D\mathbb{E}[Z|P^D\tilde{Z}]$. If $\mathbb{E}\|\mathbb{E}[Z|\tilde{Z}]\|_H^2 < \infty$, then $Z^D \to \mathbb{E}[Z|\tilde{Z}]$ in $L^2$ and almost surely.*

## C  Tables

Table 2: Summary of previous and our results for score-based diffusion models with bounded support ($B_R(0)$) target in $d$ dimensions.

| Metric | Complexity | Result |
|---|---|---|
| $\mathcal{W}_1(\overrightarrow{P}_0, \overleftarrow{Q}_{T-\delta})$ | $\log N = \mathcal{O}\left(\frac{R^2(d+R^4)^2(\log R)^2}{\epsilon_0^2}\right)$ | (De Bortoli, 2022) Thm. 1 + Cor. 2 |
| $\mathcal{W}_2(\overrightarrow{P}_0, \overleftarrow{Q}_{T-\delta})$ | $\log N = \mathcal{O}\left(\frac{R^2(d\vee R^2)^2}{\epsilon_0^4}\right)$ | (Beyler & Bach, 2025) Prop.12 |
| $\mathcal{W}_2(\overrightarrow{P}_0, \overleftarrow{Q}_{T-\delta})$ | $\log N = \mathcal{O}\left(\frac{R^2(d\vee R^2)^2}{\epsilon_0^4}\right)$ | This work: Cor. 3.12 |
| $\mathcal{W}_2(\overrightarrow{P}_0, \Pi_R\overleftarrow{Q}_{T-\delta})^{(+)}$ | $N = \mathcal{O}\left(\frac{d^3 R^8(R\vee\sqrt{d})^4}{\epsilon_0^{12}}\right)$ | (Chen et al., 2023b) Cor.5, this work Cor. D.1 |

$+$ $\Pi_R$ is the projection onto $B_R(0)$, i.e., discarding mass outside $B_R(0)$.

Table 3: Explicit expressions for the constants in the derivations

| Constant | Expression |
|---|---|
| $K$ | $\max\{1, \|AC^{-1}\|, \|A^{-1}C\|\}$ |
| $L_0$ | $K^2(\|C\nabla^2 h\|_\infty + |\sqrt{C}\nabla h|_\infty^2)$ |
| $\tilde{L}_0$ | $K^2\left(\|C\nabla^2 h\|_{\mathcal{L}(H),\infty} + 2\|\sqrt{C}\nabla h\|_{H,\infty}^2\right)$ |
| $L_1$ | $K\|C\|^{\frac{1}{2}}|\sqrt{C}\nabla h|_\infty$ |
| $\tilde{L}_1$ | $K\|C\|^{\frac{1}{2}}\|\sqrt{C}\nabla h\|_{H,\infty}$ |
| $L_2$ | $\max\{\|I - CA^{-1}\|, \|AC^{-1} - I\|\}$ |
| $M_0$ | $\max\{\mathbb{E}|\overrightarrow{X}_0|^2, \operatorname{Tr}C, 1\}$ |
| $M_2$ | $\mathbb{E}|\overrightarrow{X}_0|^2$ |
| $K_1$ | $e^{4+(e+1)\left(2(L_0+L_2)+(L_0+L_2)^2 e\tau\right)}$ |
| $K_1'$ | $e^{4+(2+b)e(1+e)B}$ |
| $\tilde{K}_1$ | $e^{4+(e+1)\left(2(\tilde{L}_0+L_2)+(\tilde{L}_0+L_2)^2 e\tau\right)}$ |
| $K_2$ | $\frac{3R^2}{(1-e^{-\delta})^2} + \frac{1}{1-e^{-\delta}}$ |
| $k_3$ | $\|C\|\|\Sigma\|^{-1}\left(\left(\|\nabla^2 G\|_\infty(|G|_\infty + |y|) + \|\nabla G\|_\infty^2\right) + \|\Sigma\|^{-1}\|\nabla G\|_\infty^2\left(|G|_\infty + |y|\right)^2\right)$ |
| $K_3$ | $e^{4+3(e+1)k_3}$ |

## D  Results under KL/TV bound

Our Gaussian tail Assumption 2 and the resulting regularity estimate Corollary 3.5 for the score function verifies Assumption 1 in Chen et al. (2023b), and their Assumption 2 and 3 are also satisfied by our Assumption 1 and 3. Thus, we can applies the Girsanov framework in Chen et al. (2023b) to obtain the KL/TV convergence bound under the Gaussian tail Assumption 2.

We follow the analysis in (Chen et al., 2023b), while modified their estimate of $\sum_{k=0}^{N-1}\int_{t_k}^{t_{k+1}}\mathbb{E}[|s_\theta(T - t_k, \overleftarrow{X}_{t_k}) - s(T - t, \overleftarrow{X}_t)|^2]dt^3$ with a decomposition aligned with the intermediate bounds we obtained in showing Theorem 3.4, detailed as follows.

---

[3]Under their notation, it appears as $\sum_{k=0}^{N-1}\mathbb{E}_{\overleftarrow{Q}_T}\int_{kh}^{(k+1)h}\|s_{T-kh}(X_{kh}) - \nabla\ln q_{T-t}(X_t)\|^2 dt$ in equation (16), within the proof of Theorem 10. For simplicity, the notations in our derivation remain the same as those in Appendix A.3.

For simplicity, we assume $C = I_d$ here aligned with the settings in (Chen et al., 2023a). Now we consider,

$$
\begin{aligned}
&\mathbb{E}[|s_\theta(T - t_k, \overleftarrow{X}_{t_k}) - s(T - t, \overleftarrow{X}_t)|^2 \\
&\lesssim \mathbb{E}[|s_\theta(T - t_k, \overleftarrow{X}_{t_k}) - s(T - t_k, \overleftarrow{X}_{t_k})|^2] + \mathbb{E}[|s(T - t_k, \overleftarrow{X}_{t_k}) - s(T - t, \overleftarrow{X}_t)|]^2 \\
&\lesssim \epsilon_k^2 + \mathbb{E}[\overleftarrow{X}_{t_k} - \overleftarrow{X}_t|^2] + \mathbb{E}[|\hat{s}(T - t_k, \overleftarrow{X}_{t_k}) - \hat{s}(T - t, \overleftarrow{X}_t)|]^2 \\
&\lesssim \epsilon_k^2 + (M_2 \vee d)(t - t_k) + (e^{\frac{t - t_k}{2}} - 1)^2 \mathbb{E}[|\hat{s}(T - t, \overleftarrow{X}_t)|^2] \\
&\quad + e^{t_k} \mathbb{E}[|e^{-\frac{t_k}{2}} \hat{s}(T - t_k, \overleftarrow{X}_{t_k}) - \hat{e}^{-\frac{t}{2}} \hat{s}(T - t, \overleftarrow{X}_t)|^2].
\end{aligned}
$$

By Corollary 3.2,

$$
\begin{aligned}
\mathbb{E}[|\hat{s}(T - t, \overleftarrow{X}_t)|^2] &\lesssim \mathbb{E}[|(I - C\bar{A}_t^{-1})\overleftarrow{X}_t|^2] + \mathbb{E}[|\tilde{s}(T - t, \overleftarrow{X}_t)|^2] \\
&\lesssim L_2^2 e^{-2t} + L_1^2 e^{-t}.
\end{aligned}
$$

With Lemma A.3 and (36) in Appendix A.3, we have

$$
\begin{aligned}
\mathbb{E}[|e^{-\frac{t_k}{2}} \hat{s}(T - t_k, \overleftarrow{X}_{t_k}) - \hat{e}^{-\frac{t}{2}} \hat{s}(T - t, \overleftarrow{X}_t)|^2] &= \mathbb{E}|\int_{t_k}^t e^{-\frac{s}{2}} \nabla \hat{s}(T - s, \overleftarrow{X}_t) \sqrt{C} d\tilde{B}_s|^2 \\
&\leq (L_0 + L_2)^2 \mathrm{Tr}(C) e^{-2T}(e^{t_{k+1}} - e^{t_k}).
\end{aligned}
$$

Subbing back, we arrive at,

$$
\mathbb{E}[|s_\theta(T - t_k, \overleftarrow{X}_{t_k}) - s(T - t, \overleftarrow{X}_t)|^2 \lesssim \epsilon_k^2 + (M_2 \vee d)\tau + (L_2^2 e^{-2t} + L_1^2 e^{-t})\tau^2 + (L_0 + L_2)^2 de^{-2T + 2t}\tau,
$$

and,

$$
\sum_{k=0}^{N-1} \int_{t_k}^{t_{k+1}} \mathbb{E}[|s_\theta(T - t_k, \overleftarrow{X}_{t_k}) - s(T - t, \overleftarrow{X}_t)|^2]dt \lesssim \epsilon^2 T + (M_2 \vee d)T\tau + (L_2^2 + L_1^2)\tau^2 + (L_0 + L_2)^2 d\tau. \quad (58)
$$

With estimate (58) replacing the first estimate in the proof of Theorem 9 of Chen et al. (2023b), we obtain the following TV bound for Gaussian tail target,

$$
\mathrm{TV}(\overrightarrow{P}_0, \overleftarrow{Q}_T) \lesssim \sqrt{\mathrm{KL}(\overrightarrow{P}_0 || \gamma_d)} e^{-T} + (\epsilon + \sqrt{(M_2 \vee d)\tau})\sqrt{T} + (L_2 + L_1)\tau + (L_0 + L_2)\sqrt{d\tau}. \quad (59)
$$

Under the smooth case where $L_0$, $L_1$, and $L_2$ are assumed to be moderate, the sampling complexity scales linearly with respect to $d$, which is suboptimal comparing to our main result Corollary 3.5. For the bounded-supported case (Assumption 5), we summarized the complexity bound as the following corollary, which is equivalent to one obtained in Corollary 6 of Chen et al. (2023b).

**Corollary D.1.** *Under the bounded $s$ Assumption 5, we have the following sampling complexity upper bound to obtain $\epsilon_0$-accuracy of $\mathcal{W}_2(\overrightarrow{P}_0, \Pi_R \overleftarrow{Q}_{T-\delta})$,*

$$
N = \mathcal{O}\left(\frac{d^3 R^8 (d \vee R^2)^2}{\epsilon_0^{12}}\right). \quad (60)
$$

*Proof.* As shown in the proof of Theorem 3.10 (Appendix A.7), the constants at early stopping time $\delta$ should take assymptotics as,

$$
M_2 = \mathcal{O}(R^2), \quad L_0 = \mathcal{O}\left(\frac{R^2}{\delta^2}\right), \quad L_1 = \mathcal{O}\left(\frac{R}{\delta}\right), \quad L_2 = \mathcal{O}\left(\frac{1}{\delta}\right).
$$

Following the proof of Corollary 6 in Chen et al. (2023b), we have,

$$
\mathcal{W}_2(\Pi_R \overleftarrow{Q}_{T-\delta}, \overrightarrow{P}_0) \lesssim R\sqrt{\mathrm{TV}(\overleftarrow{Q}_{T-\delta}, \overrightarrow{P}_\delta)} + \mathcal{W}_2(\overrightarrow{P}_\delta, \overrightarrow{P}_0).
$$

To obtain $\epsilon_0$-accuracy of $\mathcal{W}_2(\overrightarrow{P}_\delta, \overrightarrow{P}_0)$, the stopping time $\delta$ should be $\mathcal{O}(\frac{\epsilon_0^2}{\sqrt{d}(\sqrt{d}\vee R)})$. We now take $\mathrm{TV}(\overleftarrow{Q}_{T-\delta}, \overrightarrow{P}_\delta) = \mathcal{O}(\frac{\epsilon_0^2}{R^2})$ to obtain $\epsilon_0$-accuracy level for $\mathcal{W}_2(\Pi_R \overleftarrow{Q}_{T-\delta}, \overrightarrow{P}_0)$, and we also assume $T$ to be moderate. By the TV bound (59), it requires,

$$\tau = \mathcal{O}\left(\frac{\epsilon_0^4}{R^4 d(L_0 + L_2)^2}\right) = \mathcal{O}\left(\frac{\epsilon_0^{12}}{d^3 R^8 (d \vee R^2)^2}\right),$$

and the sampling complexity,

$$N = \mathcal{O}\left(\frac{1}{\tau}\right) = \mathcal{O}\left(\frac{d^3 R^8 (d \vee R^2)^2}{\epsilon_0^{12}}\right).$$

$\square$

