# OpenReview forum: "Wasserstein Bounds for generative diffusion models with Gaussian tail targets"
_TMLR — Accepted by TMLR_

### Review · Reviewer_yHjm · 2025-12-24

**Summary Of Contributions:**

This work establishes non-asymptotic bounds on the Wasserstein-2 distance for Score-based Generative Models under a Gaussian tail assumption on the target distribution. The contraction in $W_2$ is obtained by deriving global regularity estimates for the modified score function (and for the Hessian of $- \log p_t$). These estimates are obtained via heat kernel estimates applied to the Cole–Hopf transformation of the associated viscous Hamilton-Jacobi equation.

Strengths:

(1) The Gaussian tail assumption on the target data distribution allows to consider some non-log-concave data distributions for the $W_2$ analysis.

(2) The theoretical framework includes the case of spatially anisotropic noise in the forward process and infinite dimensional setting via trace-class covariance operators.

Weaknesses:

(1) The analysis relies on strong assumptions on the score network (Assumption 4), requiring uniform bounds for $\theta$ which is the neural network's parameters. This condition may be difficult to verify or justify in practice.

(2) The work is restricted to the Ornstein-Uhlenbeck process as forward process and it does not include other more general SDEs as forward processes.

(3) The comparison with existing results is limited, and relevant works are not discussed, which weakens the positioning of the main contributions.

(4) Some sentences need to be revised as they are not clear, please see section "Requested changes".

**Audience:**

Yes

**Audience Explanation:**

The paper may be of interest to the TMLR audience working on the theoretical analysis of score-based generative models. Its focus on non-asymptotic error bounds and convergence guarantees for SGMs may be of interest to researchers seeking to better understand the empirical success of these models, especially for data distributions exhibiting local non-concavity and multimodality, as covered by Assumption 2. While a more thorough comparison with recent results in the $W_2$ literature would further strengthen the paper’s positioning, the results and techniques presented are likely to be of interest to a subset of the TMLR audience working on the convergence of generative modeling.

**Broader Impact Concerns:**

N/A.

**Claims And Evidence:**

Yes

**Claims Explanation:**

The theoretical claims are supported by rigorous and technically sound proofs, and I did not identify gaps between the stated results and the provided evidence. However, while the technical correctness appears solid, the clarity of the narrative and the presentation of some arguments could be improved. In particular, certain assumptions, comparisons with prior work, and implications for convergence rates would benefit from clearer exposition and discussion. I have therefore raised several questions and requested clarifications in the "Requested Changes" section to improve the accessibility and transparency of the arguments, rather than their technical validity.

**Requested Changes:**

The references mentioned are provided at the end of this section.

Important points:

(1) Page 8. Theorem 3.3: The restriction on the step size $\tau$ is important for the convergence analysis and for practitioners. It should be clearly stated in the statement of the theorem. I would suggest adding a comparison between the restriction on $\tau$ stated in the proof at page 17 and those used in prior results, as done in Table 2 in [7]. The same comment applies to the statements of Theorem 3.9 and Theorem 3.12.

In addition, there is no discussion of the rate of convergence. Looking at prior works in $W_2$, Theorem 3.5 in [6]  and Theorem 19 in [7] achieve rate $1/2$ and $\sqrt{d}$ dependence, while Theorem 22 in [7] achieves rate $1$ and $O(d)$ data dependence. However, additional assumptions on the score network are required in order to use an explicit Milstein scheme, which allows one to obtain convergence rate one at the expense of worse data dependence, see discussion in Remark 22 in [7]. It seems to me that you manage to achieve convergence rate $1/2$ and $\sqrt{d}$ dependence at the same time because of the strong Assumption 4 on the score network, which is "kind of similar" to Assumption 3b in [7]. Could you please comment on this?

(2) Page 6, about Assumption 2: I would suggest carefully revising the sentence: "To the best of our knowledge, Assumption 2 in the form used here has not appeared explicitly in prior works on diffusion model theory". In the literature, Assumption 1 in [9] roughly state that the tail of the data distribution should decay almost as quickly as a standard Gaussian. Moreover, Assumption 2 has been used in infinite dimension in diffusion model theory in Theorem 13 in [10].

Page 7, Assumption 3: Several results in $W_2$-convergence analysis, such as [1,2,3,6,7], take the expectation with respect to the algorithm, for which the density is known; see, for instance, the discussion in Remark 3.4 in [6]. Could you please comment on your choice of taking the expectation with respect to $p_t$, which depends on the unknown $p_0$?

Page 8, Is Assumption 4 satisfied by some type of neural network? Are these requirements satisfied in practice?

(3) Page 2, Related work section: To better position the authors work within the current literature, it would be worth mentioning for unfamiliar readers that a common issue when working with the $W_2$ metric is that the backward process needs to be contractive (see, e.g., the explanation in Section 4 "Technical overview" in [8]). For this reason, earlier works require strong log-concavity of the data distribution (and sometimes other stringent regularity assumptions) to get polynomial dependence of the problem parameters, see, for example [1,2,3,4,5]. Follow-up works relaxed these strong-log concavity assumptions to weak log-concavity (together one-sided Lipschitzness of the potential) like in [6] and semiconvex target distributions with potentially discontinuous gradients in [7], and so on.

In addition, $W_2$ distance can be bounded by the TV distance under Assumption 5. A comparison between Theorem 3.9 and results in TV distance is missing.

Page 10: Corollary 5 (Section 3.2) in [8] obtains polynomial dependence in all relevant problem parameters in $W_2$, vastly improving the exponential dependencies in De Bortoli, 2023 and your Corollary 3.11. Could you please comment on this? In addition, Theorem 2.2 in [11] is also relevant for the comparison.

Page 2, Table 1: The table is incomplete and not clear. It mixes results with bounded support and unbounded support, which should be presented in separate tables.

Results with bounded support are not up to date. For instance, Corollary 5 (Section 3.2) in [8] and Theorem 2.2 in [11] mentioned above is missing. As mentioned before, the $W_2$ distance can be bounded by the TV distance. Therefore, TV-distance results should also be included in the comparison.

For the results with unbounded support, it would be helpful to distinguish between works that directly compare the law of the generative model with the data distribution and works that make use of an early stopping rule, that is, papers that compare the law of the algorithm with a smoothed version of the data (e.g. Benton et al, 2023 in the table). This distinction is important because having a KL against smoothed data distribution does not provide a $KL$ or $W_2$ bound against the original data unless further strong assumptions are made.

Assumption 1 in [1] assumes strong log-concavity and $- \log P_0$ is smooth not only log-concave as stated in the table.

(4) The work is restricted to the Ornstein-Uhlenbeck process as the forward process. Can this analysis be extended to more general SDEs, as done in [1]? If not, is this a limitation of your methodological contribution?

Minor points:

(5)  The proof of Lemma A.2 establishes (33), but there is no discussion explaining why $e^{-t/2} (s(T-t,X_t) + X_t)$ is a martingale.

(6) Page 3: I would suggest providing a reference for the following sentence: "It is well known that the function log p itself follows a viscous Hamilton-Jacobi (vHJ) equation,  as seen in (8) in the later discussion". Moreover, "the later discussion" on page 5 refers to "similar derivation are known to experts" later which is quite vague. The ML audience may not be familiar with this argument used in PDE/stochastic analysis.

Page 3: I would suggest adding references to support the sentence: "most results are seeking a spatially global Hessian bound which only lasts for a finite time".

Page 7: This statement "This result provides a formal justification for a well-known empirical feature. " also requires a reference.

(7) Page 27: In the proof of Lemma B.1, terms such as $x^{D+1:\infty}$ and $\mathcal{L}(H^D)$ are not defined in the text.

(8) Page 2, Related Work: The following sentence needs rephrasing: "In such cases, the observed data often consists of
discrete samples from an underlying function. For instance, Bayesian inverse problems (Stuart, 2010). "

(9) Page 3: Proposition 8 in Gao et al, 2023 is actually Proposition 6 (in the JMLR and ArXiv version). This issue is repeated in  Remark 3.5.

Page 30, Proposition B.4: Lemma 5 in Pidstrigach et al is actually Lemma 16. Page 22: I cannot find (A.3) from "The preliminary estimate (A.3)".

Typos and presentation issues:

(10) Pages 12-14, References: Pidstrigach et al (2023) is cited twice,  and Caffarelli (2000) contains a strange symbol. Several works have since been accepted to journals/conferences; I would suggest including this information.

(11) Page 9 and 10, "applicative scenario".  Page 15: First display $ (2 \pi)^{n/2}$, last line $|| \nabla^2 h ||_{\infty}$ and  $\mathbb{R}^n$. Page 21, after equation (36): $E(k)=\mathbb{E}|e^{t_k/2} (...) |$. Page 23: $I_2^k$ satisfies (34) rather than (36).

(12) $\hat{s}$ appears in (31) on page 17, but it is defined later on page 19. I would suggest moving the definition before equation (31).


References:

[1] X Gao, H M. Nguyen, and L Zhu. Wasserstein convergence guarantees for a general class of score-based generative models. JMLR, 2025.

[2]  S Bruno, Y Zhang, D Lim, O D Akyildiz, and S Sabanis. On diffusion based generative models and their error bounds: The log-concave case with full convergence estimates. TMLR, 2025.

[3] S Strasman, A Ocello, C Boyer, S Le Corff, and V Lemaire. An analysis of
the noise schedule for score-based generative models, TMLR, 2025.

[4] W Tang and H Zhao. Contractive diffusion probabilistic models,ArXiv, 2024

[5] Y Yu and Lu Yu. Advancing Wasserstein convergence analysis of Score-based Models: Insights from
discretization and second-order acceleration, NeurIPS, 2025

[6] M Gentiloni Silveri and A Ocello. Beyond log-concavity and score regularity: Improved convergence bounds for score-based generative models in W2-distance, ICML 2025

[7] S Bruno and S Sabanis. Wasserstein Convergence of Score-based Generative Models under Semiconvexity and Discontinuous Gradients, TMLR, 2025.

[8] S Chen, S Chewi, J Li, Y Li, A Salim, and A Zhang. Sampling is as easy as learning the score: theory for diffusion models with minimal data assumptions. ICLR, 2023

[9] F Cole, Y Lu, Score-based generative models break the curse of dimensionality in learning a family of sub-Gaussian probability distributions, ICLR 2024.

[10] Pidstrigach, Y Marzouk, S Reich, S Wang. Infinite-Dimensional Diffusion Models, JMLR, 2024.

[11]  H Lee, J Lu, Y Tan, Convergence of score-based generative modeling for general data distributions, 34th International Conference on Algorithmic Learning Theory, PMLR, 2023.

---

> ### Comment · Reviewer_yHjm · 2026-02-23
> **Follow-up revision**
>
> Thank you for updating the revised version.
>
> Related work, Contractivity of the dynamics section, Page 3. I suggest revising the following sentence regarding contractivity in $W_2$ to more accurately reflect the assumptions and results in the cited works:
>
> "For this reason, many earlier works enforce contractivity through strong structural assumptions on the data
> distribution, such as strong log-concavity or closely related regularity conditions. These assumptions ensure
> global contractivity of the reverse-time dynamics and lead to polynomial dependence of the convergence
> bounds on the problem parameters; see, for instance, (De Bortoli, 2022; Chen et al., 2023a; Benton et al.,
> 2024; Conforti et al., 2024). "
>
> Benton et al., 2024 establish bounds in KL divergence and does not assume strong log-concavity or closely related regularity conditions, see, e.g. Assumption 2.
>
> Conforti et al., 2024 works in KL bounds and does not assume strong log-concavity, see e.g. their Assumption H2 and discussion in Section 2.4: Related works and comparison with existing literature.
>
> Chen et al., 2023a similarly does not require strong log-concavity or closely related regularity conditions in the sense suggested here.
>
> If De Bortoli, 2022 is cited in this context, it may be more precise to refer specifically to Theorem 3, where a $W_1$ bound with polynomial dependence is obtained, rather than the main results, which yield exponential dependence.
>
> Overall, I recommend clarifying the metric in which the bound is established (e.g., KL vs. Wasserstein), and the precise structural assumptions used in each cited work. Alternatively, I suggest referring to works that analyze $W_2$ using strong log-concavity, such as those cited earlier in my previous comment.
>
> Thank you again for the revision.

---

> > ### Author Response · Authors · 2026-02-25
> >
> > Dear Reviewer,
> >
> > Thank you for the careful reading of our revised manuscript.
> >
> > We have revised the paragraph (in purple) to (i) explicitly distinguish Wasserstein-based analyses that rely on contractivity of the reverse-time dynamics (and cite works mentioned in the  comment that provide $W_2$ convergence guarantees under strong log-concavity or related conditions), and (ii) separately mention KL-based analyses that obtain polynomial-type dependence without assuming strong log-concavity or global contractivity, with the appropriate citations (e.g., Chen et al., 2023; Benton et al., 2024; Conforti et al., 2024).
> >
> > Regarding De Bortoli, 2022, we found that in their Theorem 3 the constants can still exhibit exponential dependence through the quantity $\Gamma$ (appearing in $D_0$), where $\Gamma$ is defined via a bound of the form
> > $$
> > \|\nabla^2 \log p_t(x)\| \le \Gamma/\sigma_t^2 .
> > $$
> > In our setting, the analogous role is played by the Lipschitz constant (e.g., $L_0$ in Corollary 3.2 and Theorem 3.3). In particular, for bounded-support targets with early stopping, $\Gamma$ scales like $R^2/\delta^2$ up to universal factors.

---

### Review · Reviewer_MUos · 2026-01-04

**Summary Of Contributions:**

The paper provides non-asymptotic guarantees in Wasserstein-2 distance for score-based diffusion models driven by an Ornstein--Uhlenbeck (OU) forward process. The generative procedure considered is the standard one: approximate the reverse-time SDE using an estimated score and implement sampling via a discrete-time Euler-type scheme. The objective is to control the distance between the data distribution $P_0$ and the law $Q_T$ of the output of the discretized reverse sampler.

\textbf{Core technical ingredient.} A central object is the modified score
$$
\tilde s(t,x) = C \nabla \log p_t(x) +C \bar A_t^{-1} x,
$$
where $p_t$ denotes the forward density and $\bar A_t$ is the OU covariance interpolation defined in the paper. Under a Gaussian-tail assumption on $p_0$, the authors prove global-in-space bounds with exponential decay in time, of the form

$$\|\nabla \tilde s(t,\cdot)\|_\infty \leq L_0 e^{-t}, \quad \|\tilde s(t,\cdot)\|_\infty \leq L_1 e^{-t/2}. $$

These estimates are used as the main stability mechanism in the analysis.

\textbf{Main $W_2$ convergence bound.} Using the above regularity, and assuming
-an $L^2$ score estimation error level $\varepsilon$, and
-a regularity condition on the learned modified score,
the paper proves a quantitative bound for the discretized reverse sampler (Theorem~3.3):
$$
W_2^2(P_0,Q_T)
\le
K_1\Big(
e^{-2T}(M_2+\mathrm{Tr}(C))
+
\varepsilon^2 T
+
\mathrm{Tr}(C)\,\tau^2
\Big),
$$
where $M_2=\mathbb{E}|X_0|^2$, $\tau$ is the maximal time step, and $K_1$ is an explicit dimension-free constant.

\textbf{Complexity consequence.} From the above estimate, choosing $T$ and $\tau$ to achieve a target accuracy $W_2(P_0,Q_T)\le \varepsilon_0$ yields an iteration complexity of the form
$$
N=O\!\left(\frac{T}{\varepsilon_0}\sqrt{\mathrm{Tr}(C)}\right)
\quad
\text{(up to logarithmic factors).}
$$

In the common case $C=I_d$, this gives an $O(\sqrt d)$ dependence, which is one of the main quantitative messages of the paper.

**Audience:**

Yes

**Audience Explanation:**

I find the paper valuable as it targets $W_2$ (not only KL/TV) and the
 exponential decay of the Lipschitz constant is a clear mechanism to prevent
error accumulation over long horizons, something that is often delicate in diffusion-model theory.
The trace-based formulation is also conceptually well-aligned with infinite-dimensional extensions.

However, the current version relies on an assumption on the learned score field that is hard to interpret
and verify in practice, and the constants/step-size constraints appear very pessimistic (especially in the
early-stopping regime). In addition, the comparison to the recent convergence literature would benefit from a
tighter "what we gain / what we pay" narrative, beyond Table~1.

**Broader Impact Concerns:**

No specific concern.

**Claims And Evidence:**

Yes

**Claims Explanation:**

To me, the proofs are clear and correct.

**Requested Changes:**

\section{Major comments}

\subsection{Assumption on the learned modified score is strong}

The main theorem assumes that the \emph{learned} modified score satisfies a global Jacobian bound at each
discretization time $(\|\nabla \tilde s_\theta(T-t_k,\cdot)\|_\infty \le L_0 e^{-T+t_k}).$

This is stronger than the conditions typically guaranteed by score matching, which control an
$L^2(P_t)$ error  but not a global $\|\nabla \cdot\|_\infty$ bound.

By contrast, many stability analyses for diffusion samplers proceed under weaker assumptions, such as
integrability of a one-sided Lipschitz quantity or other
contractivity/semiconvexity conditions. For instance, $W_2$ convergence under log-concavity or weak
log-concavity is obtained in \cite{GaoNguyenZhu2025JMLR,gentiloni2025beyond} and several KL/TV analyses with
minimal data assumptions avoid global Lipschitz control by exploiting Girsanov/chain rules for KL
\cite{ChenLeeLu2023ImprovedAnalysis,BentonDeBortoliDoucetDeligiannidis2024StochasticLocalization,ConfortiDurmusGentiloniSilveri2025SIAMMDS}.

The paper should
-Weaken the learned-field requirement:  replace $\|\nabla\cdot\|_\infty$ by a time-integrated
one-sided Lipschitz bound, or make the relaxed "integrated discrete Jacobian" condition the main assumption.
-Provide at least one checkable sufficient condition for practitioners (for example a spectrally-normalized
network class with an explicit Lipschitz constant, plus a discussion of how this interacts with approximation).

\subsection{Step-size constraints and constants must be made explicit in the main statements and discussed quantitatively}

The theorem assumes $\tau$ ``sufficiently small'' but the analysis introduces concrete restrictions
(e.g., $\tau \lesssim 1/(L_0+L_2)$ up to constants), while the complexity corollary later selects
$\tau$ to balance $\varepsilon^2T$ and $\mathrm{Tr}(C)\tau^2$.

Please state the admissible step-size condition explicitly in the main results
and discuss how it interacts with the proposed choice of $\tau$ in the complexity bound. This matters
especially for regimes where $L_0$ becomes large (notably the early-stopping/compact-support setting), because
then the admissible $\tau$ can be much smaller than the balancing choice, potentially worsening $N$.

In addition, the prefactor $K_1$ is exponential in $(L_0+L_2)$ (dimension-free but potentially huge). The paper
should add a short discussion on when these constants are expected to be moderate or not.

\subsection{Early stopping results: the exponential dependence on $R$ and $\delta$ should be clearly positioned and interpreted}

The early stopping theorem provides a bound of the same qualitative form as the main theorem but with a
prefactor $K_2(R,\delta)$ that is exponential in $R^2/(1-e^{-\delta})^2$. As $\delta\to 0$ this is extremely
large, which complicates the interpretation of the result for compactly supported or manifold-like data,
where early stopping is a standard theoretical and practical tool \cite{LyuEtAl2022EarlyStop}.

Please add a clear paragraph addressing:
-whether an exponential dependence in $(R,\delta)$ is expected to be unavoidable in $W_2$ under global
regularity control, or whether it is primarily a proof artifact;
-which step(s) in the argument generate this blow-up   and whether localized estimates could improve it (compare with "sharp Lipschitz"
approaches such as \cite{MooneyWangXinYu2024SharpLipschitz}).

\subsection{Clarify the role of the modified score vs. the usual score}

A key conceptual choice is to control $\tilde s(t,\cdot)$ rather than $s(t,\cdot)=\nabla\log p_t$. This is
 natural: as the score $s(t,\cdot)$ contains an unavoidable linear component but please add a paragraph explaining:
-why the modified score is the right object for the stability argument,
-and how this relates to score-regularity phenomena (compact vs full support) as in \cite{Stephanovitch2025RegularityOf}.

This would also clarify the modeling/training pipeline: the paper trains $s_\theta$ but assumes regularity for
$\tilde s_\theta$; the implementation path (predict $\tilde s$ directly vs reconstruct it from $s_\theta$) should
be stated explicitly.

\section{Minor comments}

-\textbf{Literature positioning:} Table~1 is helpful; I suggest adding a brief ``gain/lose'' paragraph.
For example: compared to KL/TV bounds via Girsanov,
you obtain a $W_2$ bound and a trace-based scaling, but you assume global-in-space (learned) Jacobian control.
Compared to $W_2$ results under log-concavity/weak log-concavity \cite{GaoNguyenZhu2025JMLR,gentiloni2025beyond},
you cover non-log-concave Gaussian-tail targets but pay with exponential constants.

-textbf{Exposition:} a short roadmap before the main theorem, explicitly mapping the assumptions to the error terms
(incomplete denoising / score error / discretization), would improve readability.

-\textbf{Infinite-dimensional extension:} promising, but the proof sketch should more explicitly state which
arguments remain finite-dimensional and which are replaced (compare to \cite{PidstrigachMarzoukReichWang2024JMLR}).

---

### Review · Reviewer_Wrcs · 2026-02-10

**Summary Of Contributions:**

This paper analyses the convergence of Diffusion models in the Wasserstein 2 metric under the assumption that the base distribution can be represented as $\exp(-\lVert x\rVert^2_A/2 + h(x))$ -- a perturbation of a Gaussian by a smooth potential $h$ with bounded first and second derivatives. The forward process has a constant diffusion matrix $\sqrt{C}$ that commutes with $A$.

The key contributions/strengths are the following:

1. The paper provides a convergence analysis of diffusion models under the assumption that the initial distribution is a perturbed Gaussian, which looks novel to me.

2. It shows that the score function in this case can be represented as a linear part plus a score of a distribution that can be represented as a convolution of (improper) density $\propto\exp(h(x))$ with the Gaussian noise. Thus inheriting the regularity of $h$.

3. Building on (2), the paper derives an explicit sampling-complexity bound:
   $ N(\epsilon_0) = O(\frac{\log (M_2 + \operatorname{Tr}(C)) - \log \epsilon_0}{\epsilon_0}\sqrt{\operatorname{Tr}(C)})$,
   where $\epsilon_0$ is the desired accuracy in $W_2$ and $N(\epsilon_0)$ is the number of steps required to achive it.

4. Applied to the case when $C=A=\operatorname{Id}$, it essentially shows that previously derived lower-bound $O(\sqrt{d})$ is achieved for a broader class of measures, and not limited to the Gaussian case.

Weaknesses:

1. The sampling complexity bound has a constant exponentially depending on $\lVert C\nabla h\rVert_{\infty}$ and $\lVert C\nabla^2 h\rVert_{\infty}$. While this is expected for Grönwall-type bounds, it may make the guarantee weak unless these quantities are moderate, which limits extensibility/applicability.

**Audience:**

Yes

**Audience Explanation:**

This paper would be interesting to the audience interested in the analysis of the sampling complexity of diffusion models.  It presents a class of distributions for which $O(\sqrt{d})$ convergence in $W_2$ is achievable.

**Broader Impact Concerns:**

As a purely theoretical paper, it does not raise any major concerns or ethical implications that are worth raising.

**Claims And Evidence:**

Yes

**Claims Explanation:**

I have not found errors in the proofs.

**Requested Changes:**

0. [Critical] I believe that the title is misleading. Usually, "Gaussian-type tails" means a distribution with subgaussian tails. However, the actual assumptions are much more restrictive. E.g. Mixture of two gaussians satisfy gaussian tail assumption, but does not fall into the class under consideration.

1. [Moderate] Assuming that $h$ is concave, could you compare your results with "Wasserstein Convergence Guarantees for a General Class of Score-Based Generative Models"? In which regime are your bounds sharper/looser?

2. [Moderate] I believe that the results in Section 2.3 could be achieved by an elementary proof that does involve the vHJ equation, which I will sketch below

By definition density $p_t$ is

$$
p_t(x) \propto \int e^{-||y||^2_A/2 + h(y) - \beta^{-2}_t ||x-\alpha_t y||_C^2/2}dy$$

where $\beta^2_t = 1-e^{-t}$ and $\alpha_t = e^{-t/2}$. By rearranging the term (completing the square w.r.t. $y$), we represent the integrand as $- h(y) - ||\Delta_t x-\Gamma_t y||^2_{B_t}/2 - ||x||^2_{A_t}/2$, for some matricies $\Delta_t$ and $\Gamma_t$ that can be found explicitly.

By taking $\exp(||x||^2_{A_t}/2)$ away from the integral we get
$$
\log p_t(x) = C -  ||x||^2_{A_t}/2 + \log\int e^{||\Delta_t x-\Gamma_t y||^2_{B_t}/2}e^{h(y)}dy,
$$

Finally, rescaling of $x' = \Delta_t x$ and $y'=\Gamma_t y$ will give the desired result.

---

> ### Author Response · Authors · 2026-02-10
> **Quick discussion**
>
> We’re aware of the reviewer submitted a revision of their review: we think if it’s a Gaussian mixture, it might fall into “early stopping with bounded support” which is discussed in the current submission by direct calculation. Indeed since the assumption is rather new in the complexity bound theory, we’re not completely sure how to call it. In this manuscript, we’re not pursuing a measure completely sub to Gaussian, instead we hope the tail is in fact like Gaussian (‘uniformly’ quadratic growth in negative log density form at far).
>
> Now we’re able to submit a revised version to respond the concerns of the early reviewers. We’ll do it soon with consideration to the new comments and put a notice.

---

> ### Author Response · Authors · 2026-02-14
>
> We thank the reviewer for the thoughtful comments and valuable suggestions.
> Below, we address the points raised and indicate the corresponding revisions made in the manuscript.
>
> >**Weakness:**
> The sampling complexity bound has a constant exponentially depending on...
>
> We agree that the prefactor in the sampling complexity bound depends exponentially on
> $\lVert C\nabla^2 h\rVert_\infty$ and $\lvert\sqrt{C}\nabla h\rvert_\infty$, and we clarify the origin and interpretation of this dependence.
>
> The exponential factor arises from the discrete-time stability analysis of the reverse-time flow based on a Gronwall-type argument.
> Such exponential dependence is classical in Wasserstein-based error analyses and Lipschitz change-of-variable estimates for deterministic or stochastic flows [2],[3],[4].
>
> In the absence of additional global contractivity assumptions (e.g., strong convexity), this type of dependence is generally unavoidable.
>
> Regarding the magnitude of the constants, in typical smooth-data settings $\lVert C\nabla^2 h\rVert_\infty$ and $\lvert\sqrt{C}\nabla h\rvert_\infty$ are moderate.
> By contrast, under bounded-support assumptions combined with early stopping, these quantities may deteriorate as the support radius increases or the stopping parameter decreases.
> This reflects intrinsic geometric effects of approximating sharply supported distributions via Gaussian smoothing, rather than a limitation of the analytical framework.
>
> >**Request changes (0)**
>
> We thank the reviewer for this insightful comment regarding the title and terminology.
>
> For a Gaussian mixture, we believe it may fall into the “early stopping with bounded support” setting discussed in the current submission, where the required structure can be verified by direct calculation.
> We agree that the phrase “Gaussian-type tails” may be interpreted as referring to general subgaussian distributions.
> Our assumption is more specific: rather than requiring the measure to be merely subgaussian, we impose that the tail genuinely behaves like a Gaussian, in the sense that the negative log-density exhibits a uniform quadratic growth at infinity.
>
> Since this structural assumption is relatively new in the complexity analysis of diffusion models, the terminology may not yet be standard.
> We are open to revising the title to avoid possible ambiguity and would greatly appreciate the reviewer’s suggestion for a more precise phrasing.
>
>
> >**Request changes (1)**
>
> We thank the reviewer for this suggestion.
>
> A comparison with [1] is already included in Table 1 (7th row).
> Under the additional assumption that $h$ is concave,
> their complexity bound scales as
> $N = \mathcal{O}(\frac{d}{\epsilon_0^2} \log \frac{d}{\epsilon_0})$,
> whereas our bound scales as
> $N = \mathcal{O}(\frac{\sqrt{d}}{\epsilon_0} \log \frac{d}{\epsilon_0^2})$.
> Thus, our dependence on the dimension $d$ is sharper.
> Moreover, their Proposition 6 shows that the $\mathcal{O}(\sqrt{d})$ bound is optimal when the target is the standard Gaussian, which corresponds to our setting.
>
> >**Request changes (2)**
>
> We thank the reviewer for this observation.
>
> We agree that the derivation in Section 2.3 essentially performs the same calculation described in the comment.
> In fact, the heat-kernel representation we use can be viewed as a systematic formulation of this computation after an appropriate change of variables.
>
> Our motivation for presenting it through the vHJ and heat transformation is that the heat kernel formulation makes the parabolic structure explicit and turns the
> problem into the analysis of a convolution semigroup. The gradient and Hessian estimates in Theorem 3.1
> then follow directly from the positivity and averaging properties of the kernel, yielding uniform-in-time
> bounds in a transparent way.
>
> Nevertheless, we agree that the underlying computation is elementary in nature, and we added the above discussion clarificaition in the end of the proof of Theorem 3.1.
>
> [1] X Gao, H M. Nguyen, and L Zhu. Wasserstein convergence guarantees for a general class of score-based generative models. JMLR, 2025.
>
> [2] Valentin De Bortoli. Convergence of denoising diffusion models under the manifold hypothesis. Transactions
> on Machine Learning Research, 2022.
>
> [3] Eliot Beyler and Francis Bach. Convergence of deterministic and stochastic diffusion-model samplers: A
> simple analysis in wasserstein distance. arXiv preprint arXiv:2508.03210, 2025.
>
> [4] Giovanni Brigati and Francesco Pedrotti. Heat flow, log-concavity, and lipschitz transport maps. Electronic
> Communications in Probability, 30:1–12, 2025.

---

> ### Author Response · Authors · 2026-02-15
> **Submission of Revised Manuscript**
>
> Dear AE and Reviewers,
>
> Thank you again for the valuable feedback.
>
> We have completed the revision of the manuscript, addressing the comments and incorporating the suggested changes. We look forward to your further comments and suggested changes. And we are still available to discuss and make clarifications.

---

### Decision · Action_Editor_pFvK · 2026-03-27

**Recommendation:** Accept with minor revision

**Audience:**

Yes

**Audience Explanation:**

All 3 reviewers have answered "Yes" to this question.

Comments shared by reviewers also indicate this is the case:
- "This paper would be interesting to the audience interested in the analysis of the sampling complexity of diffusion models." (Wrcs)
- "I find the paper valuable" with specifics justifying the value (MUos)
- "The paper may be of interest to the TMLR audience working on the theoretical analysis of score-based generative models." and additional comments (yHjm)

There were comments indicating a need to improve/clarify narrative and proofs, especially from MUos and also others in some other sections of the reviews. I urge the authors to revise the manuscript accordingly, taking advantage of the feedback received.

**Claims And Evidence:**

Yes

**Claims Explanation:**

All 3 reviewers have answered "Yes" to this question.

Comments shared by reviewers also indicate this is the case:
- "I have not found errors in the proofs." (Wrcs)
- "To me, the proofs are clear and correct." (MUos)
- "The theoretical claims are supported by rigorous and technically sound proofs, and I did not identify gaps between the stated results and the provided evidence." and additional comments (yHjm)

There were comments indicating a need to improve/clarify narrative and proofs, especially from yHjm and also others in some other sections of the reviews. I urge the authors to revise the manuscript accordingly, taking advantage of the feedback received.